# Distinct patterns of volcano deformation for hot and cold magmatic systems

Gregor Weber [1] ✉, Juliet Biggs [1] & Catherine Annen [2]

Volcano deformation can be detected over timescales from seconds to decades, offering valuable insights for magma dynamics. However, these signals are shaped by the long-term evolution of magmatic systems, a coupling that remains poorly understood. Here we integrate thermal models of crustal-scale magmatism with thermo-mechanical simulations of ground deformation. This allows us to determine the influence of magmatic flux over $10^5$–$10^6$ years on viscoelastic deformation spanning a 10-year observation period. Our results reveal a coupling between surface deformation and the thermal evolution of magma systems, modulated by magma flux and system lifespan. Relatively cold magma systems exhibit cycles of uplift and subsidence, while comparatively hot plumbing systems experience solely uplift. These findings align with geophysical observations from caldera systems, emphasizing the potential of surface deformation measurements as tool for deciphering the state and architecture of magmatic systems. Considering long-term magmatic system evolution is imperative for accurate interpretation of volcanic unrest.

Analyzing ground displacement in volcanic regions is crucial for assessing volcanic hazards and enhancing our ability to forecast the initiation and progression of eruptions[1,2]. Space-borne Synthetic Aperture Radar Interferometry (InSAR) has revolutionized the monitoring of volcano deformation on a global scale, offering insights into diverse patterns of volcanic deformation in both space and time[3–6]. However, temporal patterns of volcano deformation exhibit considerable variability, encompassing steady ground uplift over days to centuries (e.g., Laguna del Maule (Chile)[7]; Long Valley Caldera (USA)[8]; Corbetti (Ethiopia)[9], and alternating uplift and subsidence cycles (e.g., Campi Flegrei in Italy[10,11]; Aluto (Ethiopia)[12,13]; Askja caldera in Iceland[14]; or Okmok caldera (USA)[15,16]). Despite rapid advances in the availability of high-quality data, understanding the underlying principles governing the vast variety of ground deformation signals at different volcanoes remains a major challenge. While phases of uplift are typically attributed to emplacement of magma intrusions[3,17], a range of mechanisms to explain episodes of subsidence over timescales from months to decades have been proposed, broadly falling within four categories: (1) Pressure changes induced by fluid movement within hydrothermal systems[13,18], (2) Interactions between magma pressur-

ization and crustal heterogeniety[19], (3) Volume contraction during magma cooling, crystallization, and/or gas release[20–22], and (4) Viscoelastic relaxation of either a shell surrounding a compressible magma chamber[23–25] or of a viscoelastic layer below a pressurized reservoir[26,27].

Viscoelastic processes have been shown to markedly impact surface deformation. Source condition estimates obtained using numerical models allowing for temperature-dependent viscoelastic rheology, require lower pressures compared to analytical solutions of elastic deformation[28,29], and impact temporal trends of ground displacement on timescales of years to decades[30–33]. Conversely, numerical models investigating the evolution of magmatic systems over geological timescales suggest that, depending on the rate of magma supply, crustal thermal structures evolve over hundreds of thousands to millions of years[34–39]. Given the temperature-dependence, the viscoelastic response of rocks subjected to strain is largely governed by the long-term thermo-mechanical history of the crust[40,41]. While these timescales are much longer than those typically invoked in surface deformation modeling, we anticipate that magmatic systems undergo different styles of ground deformation at various stages of their life-

---

[1]COMET, School of Earth Sciences, University of Bristol, Bristol, UK. [2]Institute of Geophysics of the Czech Academy of Sciences, Prague, Czechia.
✉e-mail: gregor.weber@bristol.ac.uk

cycle. This temporal evolution of crustal rheology may be essential in explaining the great diversity of spatial and temporal deformation behavior that is observed at different volcanoes worldwide[42]. Yet, even state-of-the-art thermally controlled models of volcano deformation consider the temperature field and hence viscoelastic rheology to be static, often relying on simplified geometries of magma plumbing systems[25,29]. This is an attempt to reconcile the vastly different time-scales of ground movements and magmatic system evolution to explore these dependencies in space and time.

Here we use finite difference modeling to simulate the thermal evolution of the Earth's crust, driven by varying long-term magmatic fluxes (see methods section for details). This is achieved through pulsed injection of magmatic sills and dikes at three different rates (high: $4.7 \times 10^{-4}$, mid: $3.2 \times 10^{-4}$, and low: $1.9 \times 10^{-4}$ km³/yr magma flux cases) into the crust over a period of 1 million years Fig. 1). Subsequently, the temperature distribution in the crust at 200,000-year intervals is integrated into a finite element code to solve for the temperature-dependent thermo-mechanical evolution, arising from a pressure source. Viscoelastic surface deformation is then monitored over a 10-year observation period. In reality, the location and size of the source would be controlled by the mechanical conditions and history-dependent stress field[43,44], but for illustrative purposes, we use an oblate source situated at depths of either 5, 10, or 15 km and test a range of source sizes. Using this approach, we quantify the impact of long-term magma supply rates into the Earth's crust over the lifespan of a crustal magmatic system (timescales ranging from $10^5$ to $10^6$ years) on spatial and temporal patterns of surface deformation over timescales spanning $10^0$–$10^1$ years. Our primary focus is on the role of viscoelastic crustal rheology, reflecting dynamic changes in the evolving temperature structure of the plumbing system over time. We show that our model predictions are consistent with independent geophysical imaging of the subsurface and ground deformation time-series at two caldera systems in the East African Rift. Finally, we discuss the implications that can be drawn for volcano deformation studies and monitoring.

## Results and discussion
### Magma reservoir growth
To link the long-term rheological evolution of the crust to surface deformation, we first simulated the growth and thermal maturation of magma reservoirs and their surrounding wall-rocks. Our results reveal a diverse range of thermal architectures and reservoir growth conditions primarily influenced by the long-term magma supply rate and the duration of magmatic activity (Fig. 2). Initial reservoirs accumulate at near-solidus temperatures of 700 °C and gradually become hotter, reaching peak temperatures after 1 Ma of magma injection of 850–900 °C (high flux case: $4.7 \times 10^{-4}$ km³/yr), 800–850 °C (intermediate flux case: $3.2 \times 10^{-4}$ km³/yr), and 750 °C (low flux case: $1.9 \times 10^{-4}$ km³/yr). Generally, reservoir construction progresses from the bottom to the top over time, with initial growth occurring at a depth range approximately between 10 and 15 km, where ambient crustal temperatures are higher compared to shallower levels. In each magma flux scenario, as time progresses, the isotherms within the upper 10 km of the model domain gradually rise towards the surface, with the steepest gradient observed in the high magma flux case. Simultaneously, temperature distributions become more uniform in regions deeper than 10 km and more concentric within the growing reservoirs through time. However, strong local disturbances in the temperature field result from the last few magma pulses.

As magmatic dikes and sills inject the crust in random locations, the evolving crustal temperature field is subjected to stochastic effects. A repeated run of the intermediate magma flux case demonstrates similar reservoir growth conditions, with a persistent magma body being emplaced between 200 and 400 ka since the onset of magma injection (Figs. 2f–j; 3a–e; Fig. S1). However, the fine-scale temperature distribution is mostly influenced by the spatial history of magma injections, particularly in the earlier stages of the magmatic system.

Melt fraction plots indicate that during the early stages, the magmatic system comprises multiple pockets that coalesce into a

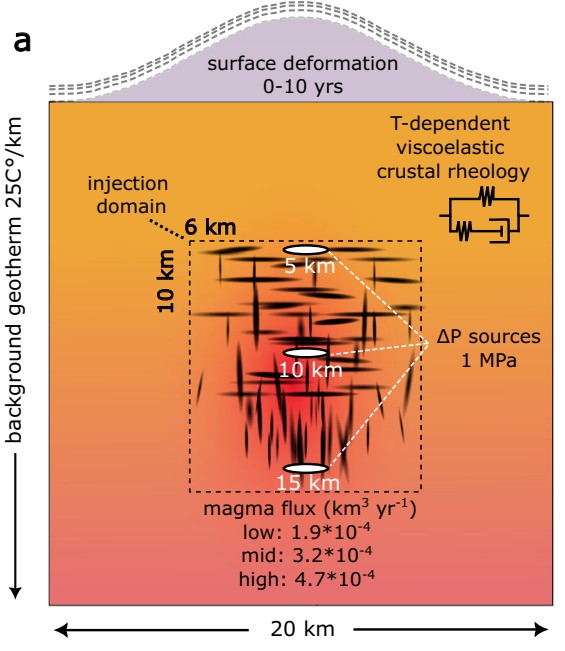

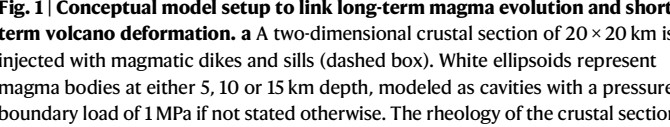

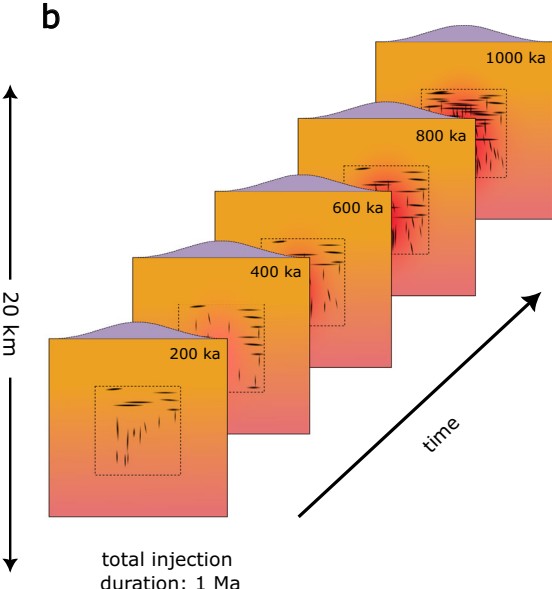

**Fig. 1 | Conceptual model setup to link long-term magma evolution and short-term volcano deformation. a** A two-dimensional crustal section of 20 × 20 km is injected with magmatic dikes and sills (dashed box). White ellipsoids represent magma bodies at either 5, 10 or 15 km depth, modeled as cavities with a pressure boundary load of 1 MPa if not stated otherwise. The rheology of the crustal section is parameterized as a temperature dependent viscoelastic model. Surface deformation in response to the crustal overpressure sources is recorded in 6-month time steps for 10 years. **b** The crustal temperature field resulting from pulsed dike and sill injection is tracked every 200 ka up to 1 Ma and subsequently used for deformation modeling.

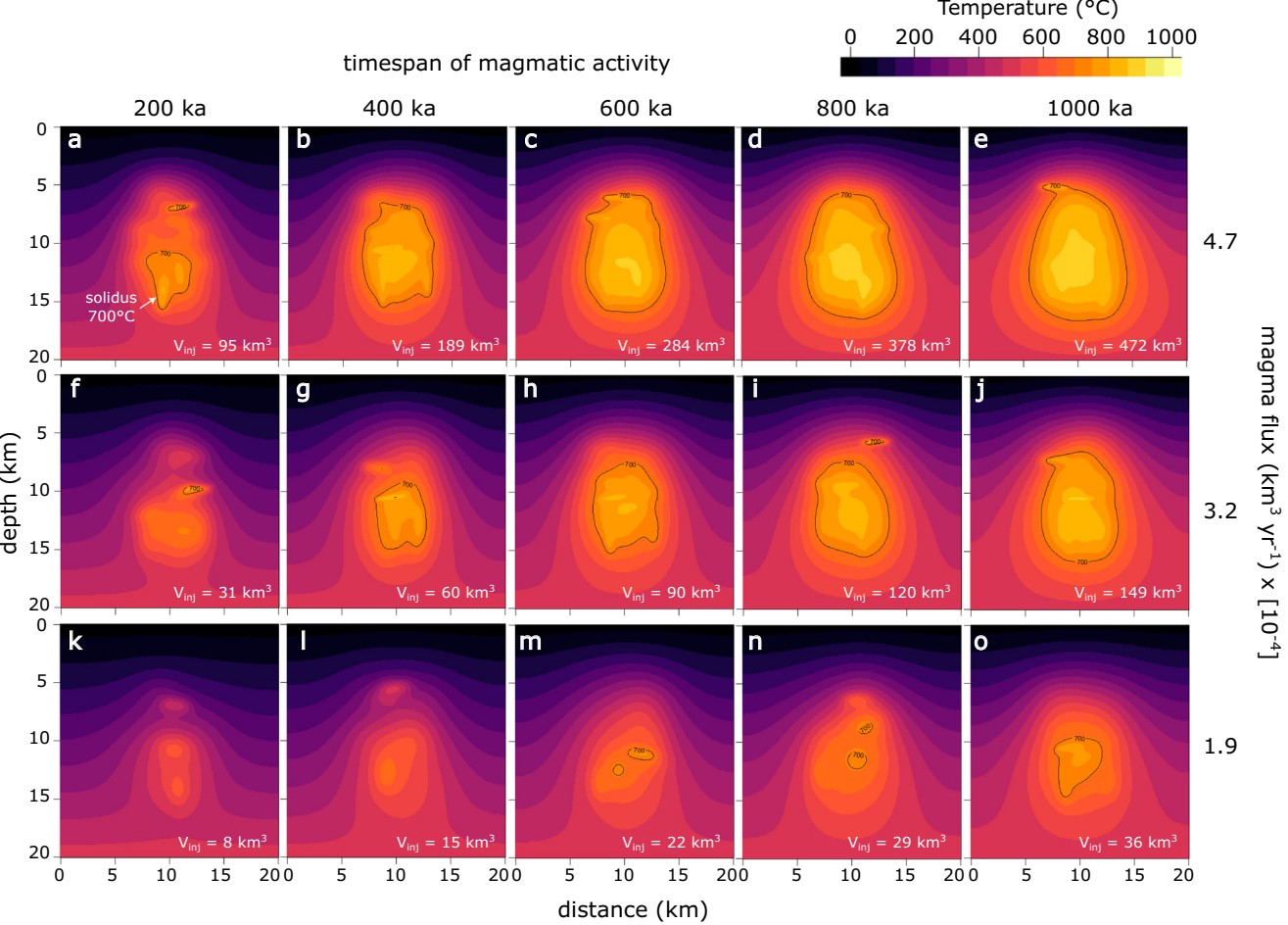

**Fig. 2 | Changes of crustal temperature field with timespan of magmatic activity.** Each panel shows the crustal temperature field (°C) at different evolutionary stages of the magmatic system, with snapshots at 200 ka, 400 ka, 600 ka, 800 ka and 1000 ka. Total injected magma volumes ($V_{inj}$) and the 700 °C isotherm (solidus temperature) are indicated on each plot. **a–e** Temporal evolution of a magmatic system built with a magma flux of $4.7 \times 10^{-4}$ km³/yr ('high flux scenario'). **f–j** Crustal temperature variation with time for a magma supply rate of $3.2 \times 10^{-4}$ km³/yr ('medium flux scenario'). **k–o** Temperature evolution of a magmatic flux of $1.9 \times 10^{-4}$ km³/yr ('low flux scenario').

single reservoir in later stages (Fig. 3f–j). In each case, early-stage reservoir growth is characterized by crystal mush with melt fractions <50%, increasing towards liquid-rich (i.e., eruptible) reservoir cores with about 75% melt in the high flux case, and 50% for the intermediate flux case after 1 Ma of magmatism. As shown by various modeling studies, volatile exsolution can modulate the pressurization history of magma reservoirs[45,46], and deforming crystal mushes may be parameterized as poroelastic or poro-viscoelastic material[47–49]. Here we focus solely on thermo-viscoelastic effects considering that sensitivity analyses indicate their dominant contribution to surface deformation[49] and given that the distribution and evolution of porosity in magmatic mush systems in not well understood.

The viscosity structure of the crust evolves in tandem with changes in temperature and melt fraction fields (Fig. 3k–o). Across the timeframe of magmatism from 200 ka to 1 Ma, crustal viscosities change in the low magma flux scenario from $10^{25}$ to $10^{20}$ Pa s at 5 km depth, $10^{18}$ to $10^{17}$ Pa s at 10 km, and $10^{19}$ to $10^{17}$ Pa s at 15 km depth (all depths at the center of the model domain). Over the same period, the intermediate magma flux case results in viscosities of $10^{18}$ to $10^{16}$ Pa s at 5 km depth and $10^{16}$ to $10^{15}$ Pa s at 10 and 15 km depth. In the high magma flux case, crustal viscosities evolve from $10^{17}$ to $10^{15}$ Pa s at 5 km depth, $10^{16}$ to $10^{15}$ Pa s at 10 km, and remain on the order of $10^{15}$ Pa s at 15 km depth. Viscosities around $10^{15}$ Pa s typically result in viscoelastic relaxation times (calculated as: $\tau \sim \eta/(0.5 \ast G)$, where $\tau$ is the relaxation

timescale in s, $\eta$ is the viscosity in Pa s, and $G$ is the shear modulus in Pa) of a few days or less, rendering them observable only with very high temporal resolution methods. Conversely, viscosities approximately at $10^{20}$ Pa s correspond to relaxation timescales spanning hundreds of years, suggesting effectively elastic behavior over typical observation timescales. Viscosities falling within the intermediate range of $10^{17}$–$10^{18}$ Pa s, however, exhibit relaxation timescales on the order of a few years, impacting our observations of volcano deformation. These considerations show that long-term magma flux and duration of magmatism exert first order controls on crustal rheologies and are therefore expected to give rise to differences in surface deformation patterns.

**Long-term evolution of volcano deformation**

We model short-term deformation patterns over a time span of 10 years for overpressure sources at 5, 10 or 15 km depth and at different long-term evolutionary stages of magmatic systems (i.e., from 0 to 1 Ma in 200 ka increments; cf. Figure 1). The predicted vertical displacements exhibit distinct temporal patterns influenced by the duration and magma flux of the long-term magmatism, and the depth of the overpressure source Fig. 4). The absolute values of displacement produced in these models scale with the source overpressure applied and are overestimated by the 2-D model set-up. Similarly, cavity size has an impact on the magnitude of displacement, but not the overall

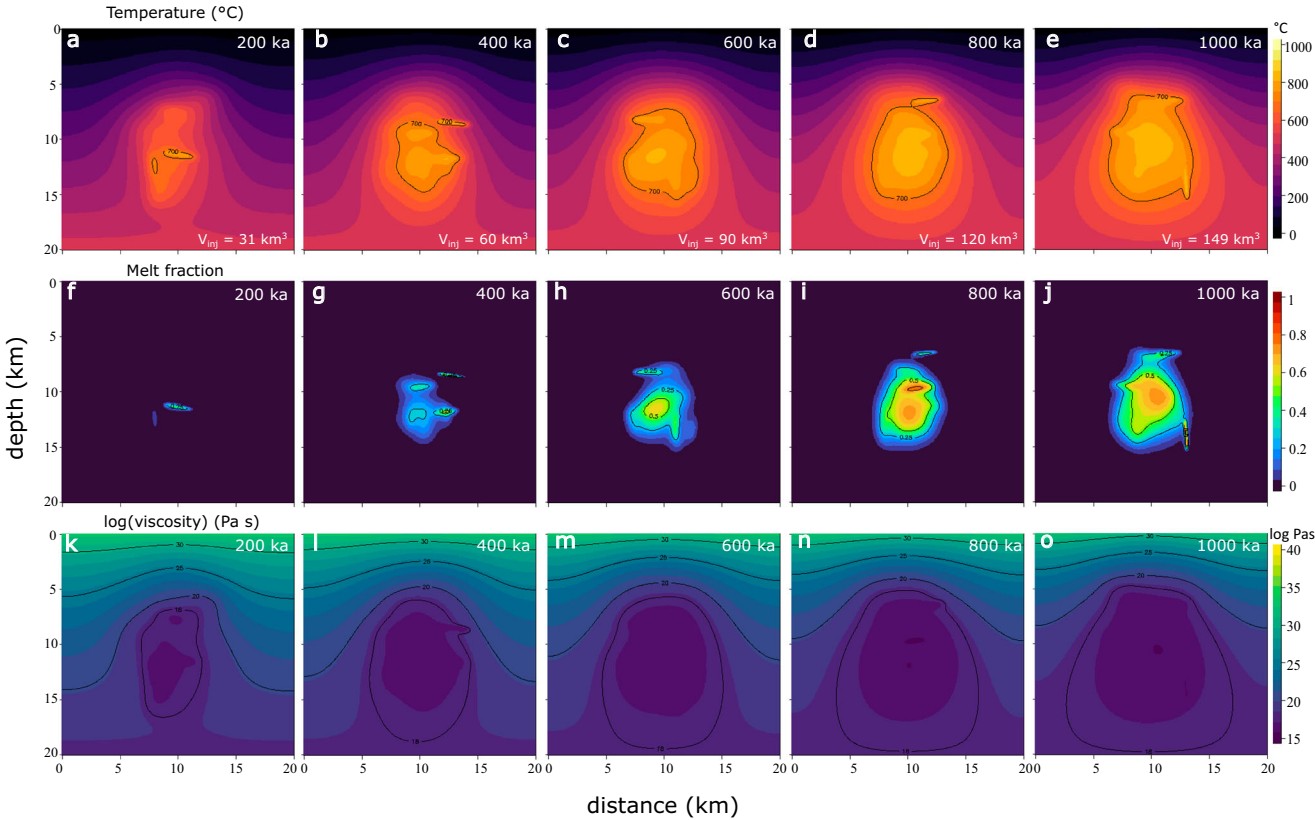

**Fig. 3 | Evolution of temperature, melt fraction, and viscosity.** Results are shown for a medium flux scenario with a magma flux of $3.2 \times 10^{-4}$ km³/yr. Timespan since the onset of magma injection is indicated each plot. **a–e** Temperature evolution between 200 and 1000 ka. Black solid lines mark the 700 °C isotherms. Total injected magma volumes ($V_{inj}$) are shown on each subplot. **f–j** Evolution of melt fraction with time. Black lines indicate constant melt fraction of 0.25, 0.5 or 0.75. **k–o** Temporal evolution of logarithmic crustal viscosity in Pa s. Black isolines mark log viscosities of 15, 20, 25, and 30.

pattern (Fig. S7). Thus, we focus on the temporal patterns and time-scales of displacement rather than absolute values.

Ultimately, the elastic or viscous response of crustal materials to deformation is governed by the ratio of relaxation to forcing time-scales (Deborah Number)[43,44]. Although the relaxation behavior in our model is governed by distribution of viscosities rather than single values, we do not model the injection process itself. The step-change cavity pressurization does not reflect a specific igneous process and cannot be considered equivalent to a forcing timescale due to magma injection. Instead, our study focuses on the time-dependent thermo-viscoelastic behavior of crustal rocks in response to a pre-formed, pressurized cavity.

The viscoelastic deformation history of sources at 10–15 km depth generally follows a similar trajectory: an initial rapid uplift, typically spanning 6 months to 2 years, succeeded by slower deformation. Given that magma intrusion pulses are expected to operate over timescales of days to years in most volcanic systems[50], these relatively short timescales suggest that discerning surface displacement result-ing from magmatic intrusion and subsequent viscoelastic deformation may pose challenges, particularly for deep sources, as also noted in previous studies[29,51]. This behavior remains consistent across various magma supply rates, although during the initial stages of magmatism (<400 ka), the transition between these two regimes extends over several years for low and intermediate fluxes.

In contrast, pressure sources at 5 km depth produce both uplift and subsidence (Fig. 4) with notable differences influenced by the thermal state and the rheology of the crust, which depend on the magmatic flux, the history of injection locations, and duration of magmatism (Fig. 3). The overall magnitude of surface displacement generally increases with the magma supply rate, showing peak values in scenarios featuring high magma flux, while cases with reduced magma flux result in lower displacements. For 200 ka of magma injection at low and intermediate magma supply rates, subsidence is recorded for the entire 10-year observation period, whereas, in the high magma flux scenario, initial subsidence is only observed in the first year. From 400 ka to 1 Ma, in the high magma flux scenario, uplift occurs at gradually decreasing rates. Conversely, intermediate flux scenario initially experiences a subsidence phase spanning up to 600 ka. At 1 Ma of magma injection, in the low flux scenario, sub-sidence is initially observed followed by uplift. Notably, the transition point from subsidence to uplift occurs progressively earlier and the total subsidence increases with longer durations of magmatism Fig. 5).

As illustrated by a repeat run of the intermediate flux scenario (dashed lines in Figs. 4, 5b), the magnitude and temporal evolution of surface deformation can be influenced by the long-term history of injection locations, which are randomized in our modeling framework. Although the overall temporal trends are similar, the spatial injection history can significantly affect the magnitude and duration of viscoe-lastic subsidence and uplift patterns. More spatially focused magma injection can lead to higher local temperatures and, consequently, greater magnitudes of viscoelastic deformation (e.g., intermediate flux 800 and 1000 ka repeat runs; Fig. 5b). Therefore, the history of spatial injection locations may be an important parameter in explaining the diversity of deformation signals that occur globally.

Although, temporal patterns of surface deformation are impacted by long-term magmatic processes, coupling thermal and thermo-mechanical simulations reveals that spatial pattern of surface defor-mation is primarily influenced by source depth and geometry rather than the thermal architecture and growth conditions of the magmatic reservoir (Fig. S2). The initial, elastic response of the crust to the

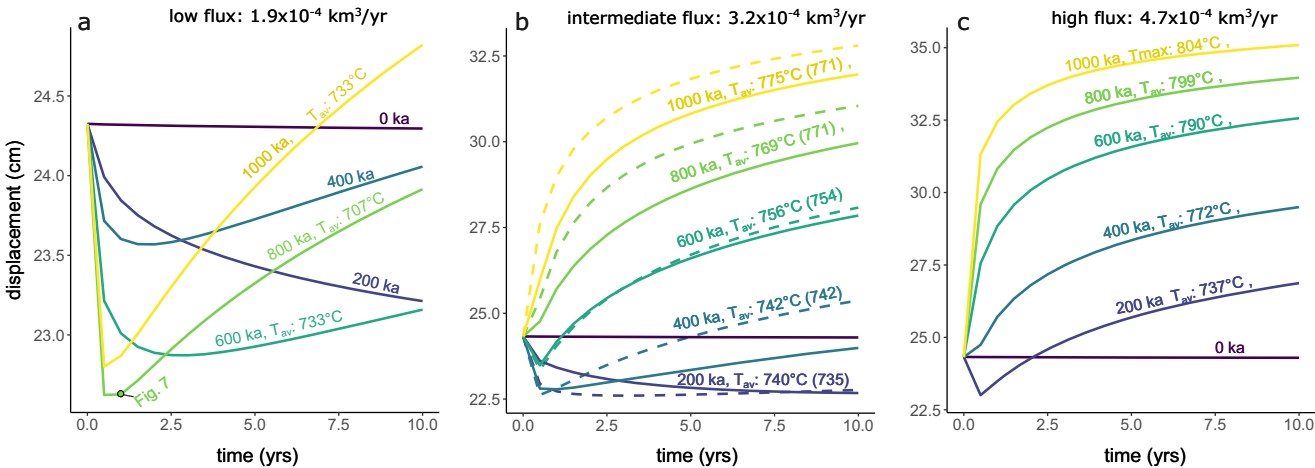

**Fig. 4 | Impact of magmatism duration, magma flux, and overpressure source depth on vertical displacement time-series for a 800 m × 200 m source.** Each panel shows the time evolution of the maximum vertical displacement over 10 years. The duration of magmatism increases from 200 ka to 1000 ka from the left to the right (cf. Fig. 1). Overpressure source depth increases from 5 km to 15 km from top to bottom. Results are shown for the high magma flux scenario (purple), intermediate flux (orange) and low magma flux scenario (red). The dotted brown curve represents a repeat run of the intermediate magma flux with different random combination of injection sites. The initial elastic response is shown as black vertical line at $t = 0$. Note that the initial elastic response in vertical displacement starts from 0 cm in each panel, which is not shown. The impact of cavity size is shown in Fig. S7.

**Fig. 5 | Viscoelastic subsidence and uplift pattern.** Each curve shows the maximum vertical displacement in cm versus time (yrs) at different temporal stages (200:1000 ka) of the magmatic system. The overpressure source was located at 5 km depth for all shown simulations. Labels reflect the average temperature of magma above solidus and the duration of magmatic activity in the numerical simulation. **a** Low magma flux scenario ($1.9 \times 10^{-4}$ km³/yr). **b** intermediate magma flux scenario ($3.2 \times 10^{-4}$ km³/yr). Dashed lines show displacement pattern of the repeat run with average temperature in parentheses. **c** High magma flux scenario ($4.7 \times 10^{-4}$ km³/yr).

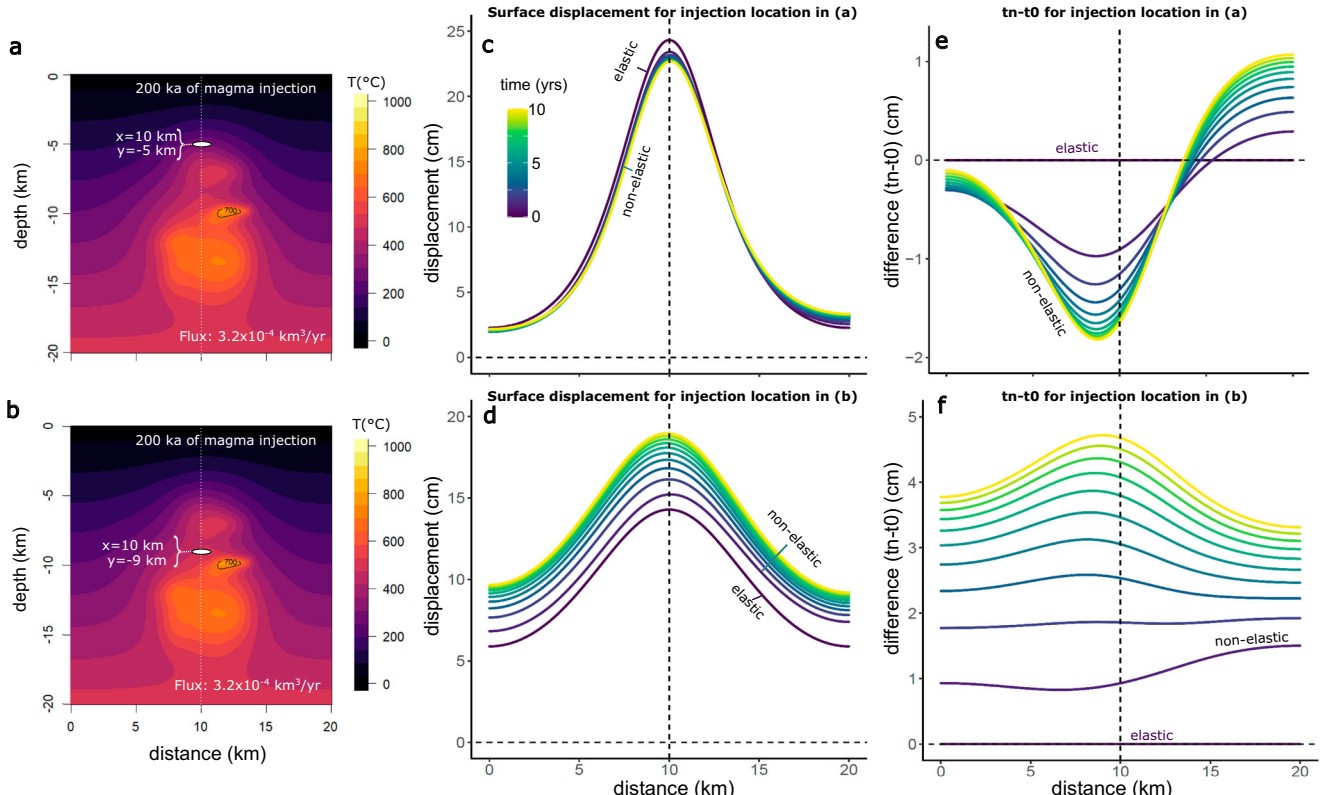

**Fig. 6 | Influence of thermal heterogeneity on spatial surface displacement pattern. a** Temperature field (°C) after 200 ka of magma injection for the intermediate flux scenario. White ellipsoid marks the overpressure source at 5 km depth with central x-coordinate at 10 km. **b** Temperature field (°C) after 200 ka for the intermediate flux scenario with pressure source centered at 10 km in the horizontal distance and depth of 9 km. The white dashed line in (**a**, **b**) marks the center of the overpressure source. Note the non-uniform distribution of isotherms in (**a**, **b**). **c** Vertical displacement (cm) versus distance (km) for the simulation shown in (**a**). The time dependent viscoelastic deformation pattern shows asymmetric subsidence relative to the symmetrical elastic response. Color coding reflects the deformation time in years. **d** Displacement pattern for the simulation shown in b, developing an asymmetrical uplift pattern over time. **e**, **f** Corresponding difference of visco-elastic displacement ($t_n$) relative to the elastic displacement ($t = 0$; horizontal line).

pressurized cavity is insensitive to the thermal structure, but the spatial pattern of the viscoelastic response is affected by thermal heterogeneities Fig. 6). Thermal structures in which isotherms are irregularly distributed relative to the overpressure source results in vertical displacement patterns that develop asymmetry over time in the visco-elastic phase. In the specific cases (Fig. 6a, b), where higher temperatures prevail on the right-hand side relative to the pressure source, displacement patterns exhibit spatial differences in the extent of viscoelastic creep, evolving over time (Fig. 6c, d). This leads to asymmetry in visco-elastic displacement, relative to the elastic component of displacement at $t = 0$ years, with a shifted center of displacement and varying magnitude response on the sides of the model domain (Fig. 6e, f). Given that the source in both scenarios remains constantly pressurized at 1 MPa, the shift can be attributed to heterogeneous viscoelastic deformation.

To summarize, viscoelastic displacement over an observation period of 10 years is influenced by the duration of magmatism, magma flux, heterogeneity of the thermal structure, and overpressure source depth. Longer durations of magmatism and higher magma fluxes generally result in greater displacement, although deeper overpressure sources exhibit similar displacement for different magma fluxes. Shallow overpressure sources show initial subsidence in some cases, particularly for shorter durations of magmatism and lower magma fluxes. This subsidence is not observed in deeper source depths. Lastly, asymmetries in the surface deformation pattern may develop over time in response to heterogeneous viscoelastic structure.

## Origin of viscoelastic subsidence

The observation that several simulations with a pressure source at 5 km depth exhibit subsidence over a wide range of timescales and magnitudes is unexpected, given previous studies indicating that uplift should predominate when a constant pressure (stress) boundary load is applied[28,32,33,52–54]. We use the low flux case, with a pressure source at 5 km and 800 ka of magmatism to illustrate the origin of this subsidence Fig. 7). The initial elastic deformation field at $t = 0$ years displays a pronounced upwards component and a smaller downwards-directed vector field, resulting in net uplift. One year after the initial timestep, the downwards component has markedly increased in magnitude, caused by rapid viscoelastic creep towards the high-temperature zone underlying the shallow overpressure source (Fig. 7b). This effect leads to a net ground displacement at the surface lower than the initial elastic surface uplift, manifesting as subsidence. Although viscoelastic creep towards the high-temperature zone beneath is still ongoing after 5 years, the process has now slowed (Fig. 7c). At this stage, the greater magnitude of the upward-directed and much slower viscoelastic creep results in ground uplift relative to the initial state.

The duration and magnitude of the subsidence period, as well as the timing of the transition from subsidence to uplift, depend on the thermal state of the plumbing system and the relative location of the overpressure source. If ambient temperatures are high, such as in the high flux case and at later magmatic stages for the intermediate flux, the effect is not observed (Fig. 4). This is because temperatures surrounding the source are high in all directions, leading to a similar

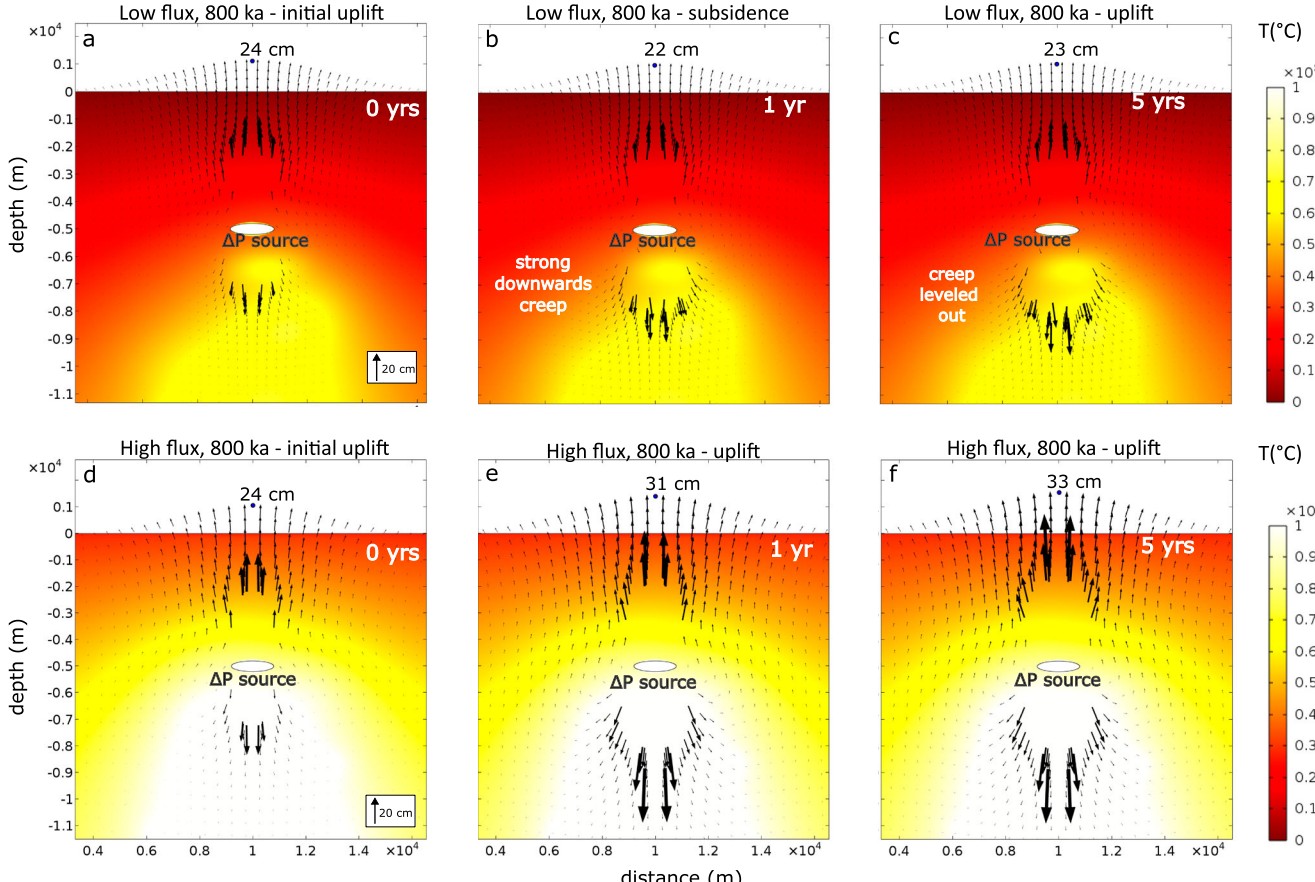

**Fig. 7 | Evolution of the viscoelastic deformation field.** Note the evolution of downwards directed viscoelastic creep towards high temperature zones. Each panel shows the temperature field (°C) after 800 ka of magma injection for the low magma flux case (**a–c**) and high magma flux scenario (**d–f**). Arrows represent the deformation field in response to an overpressure boundary load at the white ellipsoidal magma body. **a** shows the initial elastic response. **b** Deformation field after 1 year, resulting in subsidence. **c** Deformation field after 5 years. **d** Initial deformation field for the high flux scenario. **e** Deformation field after 1 year, and **f** after 5 years. The blue dot in each panel marks the maximum vertical displacement.

viscoelastic response timescale in the upwards and downwards directions. However, if ambient temperatures surrounding the overpressure source are low and much higher in the underlying plumbing system, subsidence is observed. The duration of this process, and therefore the duration of subsidence, is thus governed by the timescale of viscoelastic creep.The occurrence of either subsidence or uplift can be predicted by analysing the difference in relaxation timescales of crustal rocks, which is primarily governed by their viscosity structure. When there is a significant difference in relaxation timescales between the rocks above and below the pressure source, subsidence is likely, as the upper layer behaves elastically while viscous relaxation occurs in the lower layer (see Fig. 8). In contrast, when the difference is minimal, both layers undergo time-dependent deformation, resulting in net uplift. Thus, the relative difference in relaxation timescales, rather than absolute values, is the key factor controlling the deformation behavior.

In this study, we do not assume any specific mechanism for pressure generation within the modeled cavity. However, two widely accepted processes for magma pressurization are the injection of fresh magma into a preexisting reservoir via a feeder dike and the accumulation of an exsolved volatile phase within the magma body[50]. For volatile-driven mechanisms, where pressure is generated internally, we do not expect these processes to directly influence the relaxation behavior of the crustal rocks. However, magma transport through a feeder dike system could affect the stress field beneath the intrusion, potentially counteracting the downward-directed flow of hot host rocks. More complex models are needed to explore the interaction between magma injection and viscoelastic crustal rheology to determine whether this hypothetical effect would be significant. Nonetheless, petrologically constrained magma recharge timescales, typically ranging from days to months[50], are shorter than the 10-year observation window modeled in this study. Therefore, our results are applicable to scenarios in which the injection process has terminated.

Our results are broadly consistent with the mechanism suggested by Yamasaki et al.[26] to explain post-elastic subsidence, namely, downwards-directed viscoelastic creep resulting from a heterogeneous thermo-mechanical architecture of the crust. However, while Yamasaki et al.[26] focus on a two-layer model where injections are confined to the boundary between an elastic and viscoelastic crustal layer, we account for different rheological architectures of the crust resulting from different magma supply rates and durations of magmatic activity. This allows to map out under which conditions of reservoir growth and magma system lifespan such effects are expected to prevail. Downwards-directed viscoelastic creep is distinct from subsidence mechanisms found in viscoelastic shell models[23–25], which show subsidence only in calculations with high magma compressibility and short magma injection durations compared to the viscoelastic relaxation timescale. These models do not account for viscoelastic creep in different spatial dimensions. Our results show that post-elastic subsidence can occur independent of the duration of the injection episode and independent of magma compressibility. However, in accordance with the mechanism proposed by Townsend[25], post-elastic

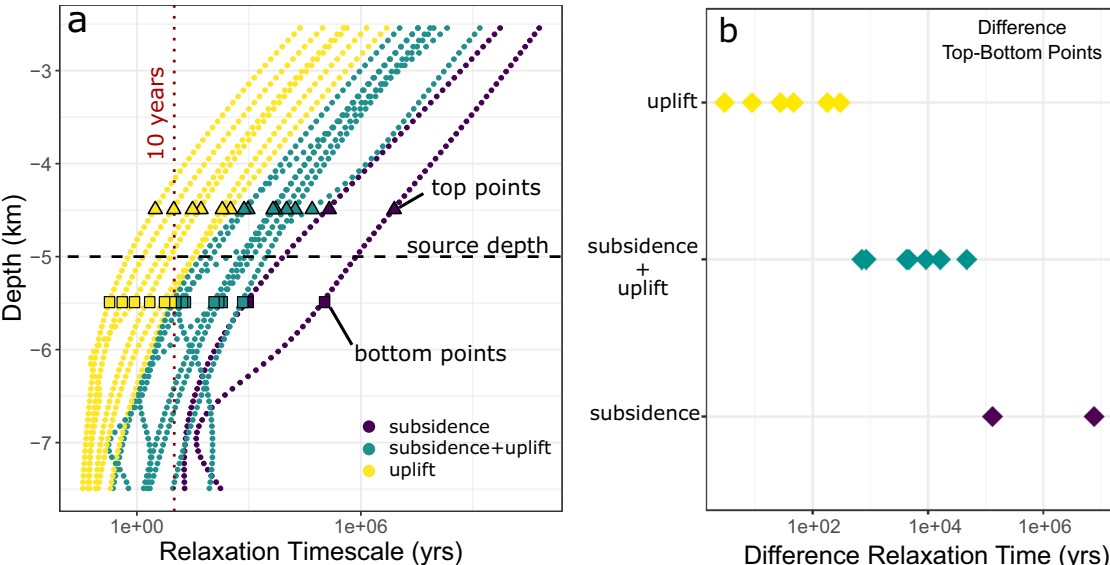

**Fig. 8 | Viscoelastic relaxation timescale in relation to deformation mechanism. a** Relaxation timescale (η/μG) with depth (km). The depth of the pressurized cavity is marked by the black dashed line. Selected points, 500 m below and above the pressure source are shown by squares and triangles, respectively. Color coding reflects deformation behavior: subsidence (purple), subsidence+uplift (green), and uplift (yellow). **b** Differences between top and bottom points in relaxation time.

subsidence does not necessarily involve a volume change of the magma reservoir.

### Comparison to geophysical observations

While the particular mechanism generating surface deformation is often difficult to discern at volcanoes and not seldom fiercely debated, our model produces testable predictions on the relationship of temporal surface deformation patterns and the thermal architecture of magmatic feeding systems that can be verified against real-world observations. Our results define distinct visco-elastic phases of subsidence followed by uplift in relatively cold magmatic systems, contrasting with hot systems exhibiting a singular pattern of ground uplift (see Figs. 4, 5; Table S1). We test these model predictions against the findings of recent geophysical imaging of the igneous feeding systems and the observed surface deformation patterns at two neighboring East African volcanoes: The Aluto and Tulu Moye volcanic complexes.

Both Aluto and Tulu Moye are large, nested caldera complexes, located approximately 50 km apart in the Central Main East African Rift[55–57]. They share several prominent features, such as bimodal basalt-rhyolite magma suites, giving rise to large magnitude explosive eruptions and smaller effusive events[13,58]. Both volcanoes have extensive eruptive histories, spanning at >300 ka for Aluto[13] and >100 ka for Tulu Moye[59]. The most recent activity at Aluto, marked by an explosive event, occurred about 400 years ago[13] and historical accounts and eyewitness reports suggest emplacement of a rhyolitic lava flow at Tulu Moye around the year 1900 CE[57,60]. Despite these similarities, recent 3D magnetotelluric imaging has revealed contrasting states of the magmatic plumbing system beneath the two volcanoes[61].

The magnetotelluric method images the electrical conductivity structure of the subsurface based on natural electromagnetic field variations. Anomalies of higher electrical conductivity beneath volcanoes can be attributed to variations in temperature, the presence of melt, saline geothermal fluids, and/or hydrothermal alteration minerals[62,63]. At Aluto and Tulu Moye, an overall similar conductivity structure has been imaged Fig. 9; refs. 61,64,65.). This trifold architectural pattern, consisting of a shallow-level anomaly interpreted as a hydrothermal alteration cap, underlaid by intermediate and deep-level conductivity anomalies, is a distinctive feature observed in numerous volcanoes worldwide[63]. However, at Aluto and Tulu Moye, significant differences exist in the magnitude of the intermediate (~4–6 km depth) and deep (>8 km depth) conductivity anomalies, showing higher conductivity values for Tulu Moye.

Combining petrological phase equilibria and electrical conductivity modeling, these differences in conductivity can be attributed to temperature and melt fraction differences for the anomalies beneath each volcano[61]. At Aluto, temperatures of 725–745 °C have been inferred, while modeling the magnetotelluric signal at Tulu Moye indicates temperatures between 785–905 °C. As shown in Fig. 10 and Table S2, these temperature ranges are well represented by the temperature distributions of our simulations. We suggest that average temperatures of the magma above solidus are most relevant to the comparison given transient effects that impact extreme values, as well as considering the resolution of magnetotelluric imaging on spatial scales of ~$10^2$–$10^3$ m and the consequent averaging of the temperature distributions. Generally, model runs with average temperatures within the range observed at Aluto show either subsidence or subsidence that transitions into uplift at a later stage, while simulations with mean temperatures corresponding to Tulu Moye show monotonous uplift (Fig. 10; Table S2; Fig. 5). Matching of our predictions and temperature distributions for Aluto and Tulu Moye thus suggests that these differences could be driven by either contrasting durations of magmatic activity for the two volcanoes at the same long-term magma input rate (e.g., high flux scenario 200 ka versus 800 ka; Fig. 10; Table S2), or reflect magmatic systems at a similar evolutionary stage that have been constructed with contrasting long-term magma addition rates (e.g., low flux scenario versus high flux case at 600 ka). Long-term crustal magma flux estimates at Tulu Moye and Aluto are, however, not available and published geochronological data does not permit us to distinguish between the different possibilities.

Surface deformation has been detected using InSAR at both Tulu Moye and Aluto (Biggs et al. 2011). Between 2004 and 2010, Aluto showed two cycles of uplift lasting several months, each followed by longer periods of subsidence (Fig. 9). In contrast, no subsidence has been observed at Tulu Moye, but solely ground uplift has in the period between 2008 and 2010, as well as since 2016 (refs.[12,66]; COMET Volcano Deformation Portal: https://comet.nerc.ac.uk/comet-volcano-portal/; Fig. 9). Estimates of the deformation source depth and geometry for both volcanoes were carried out using elastic analytical

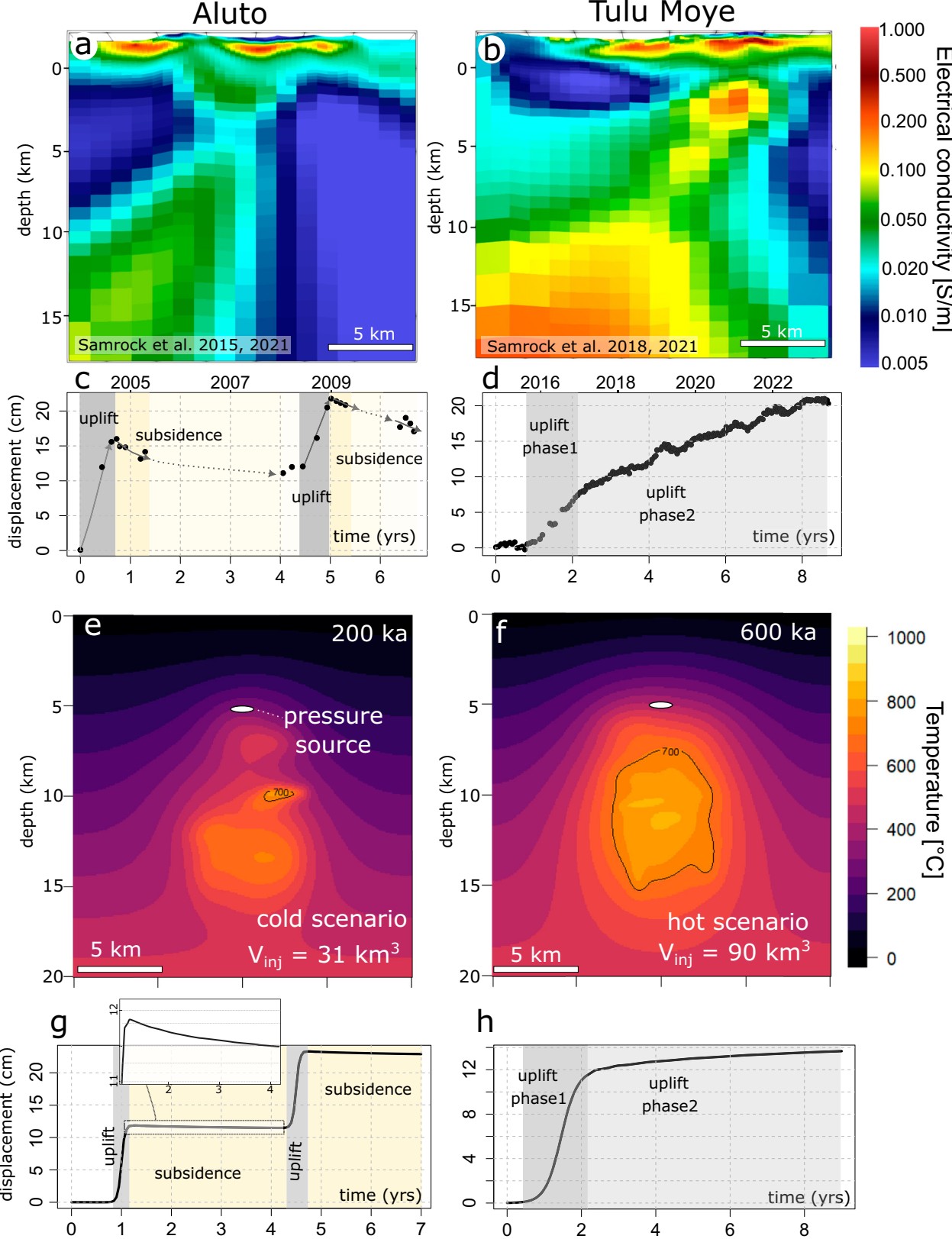

solutions. The deformation pattern at Tulu Moye can be explained by a sill-like geometry at about 7.7 km depth beneath the surface[67] and at Aluto by a point source at a depth range between 4.8 and 5.4 km[13].

These observations are fully consistent with our modeling results suggesting continuous uplift pattern for hot plumbing systems like Tulu Moye and subsidence-uplift cycles for colder magmatic systems

like Aluto (Figs. 8, 9; Table S2). Comparing the deformation source estimates to the thermal structure of the plumbing system beneath the two volcanoes reveals that both pressure sources are underlain by a high temperature zone but also that the ambient temperatures at the level of the pressure source are higher in the case of Tulu Moye. In Fig. 9, we present two simulations: one for Aluto (intermediate flux,

**Fig. 9 | Comparison of modeling results with thermal state and deformation patterns for East African Rift volcanes. a, b** Magnetotelluric imaging of Aluto[61,65] (Samrock et al. 2015, 2021; CC BY-NC-ND) and Tulu Moye volcano[61,64], highlighting higher electrical conductivity (S/m) in warmer colors. Higher electrical conductivities have been modeled as indicative of increased temperature and melt fraction[61]. Reprinted from Samrock, F., Grayver, A. V., Bachmann, O., Karakas, Ö., & Saar, M. O. (2021). Integrated magnetotelluric and petrological analysis of felsic magma reservoirs: Insights from Ethiopian rift volcanes. Earth and Planetary Science Letters, 559, 116765. Copyright © 2024, with permission from Elsevier. **c, d** Surface displacement observations, revealing uplift (dark shading) and subsidence (bright shading) cycles at Aluto[12], and monotonous but variable slope uplift

at Tulu Moye (COMET Volcano Deformation Web Portal: https://comet.nerc.ac.uk/comet-volcano-portal/). **e, f** Temperature fields post 200 ka and 800 ka of magma injection, respectively, displaying peak temperatures consistent with temperature reconstructions based on magnetotelluric data. White ellipsoid denotes the 5 km depth overpressure source. **g** Surface deformation simulation for 'cold' scenario **e** modeled with two consecutive pulses of overpressure (Fig. S3), resulting uplift-subsidence cycles. The inset provides a close-up view of a subsidence episode. **h** Displacement time-series for 'hot' scenario **f**, modeled as a single overpressure pulse, showing an initially steep phases of uplift, followed by uplift with gentler slope.

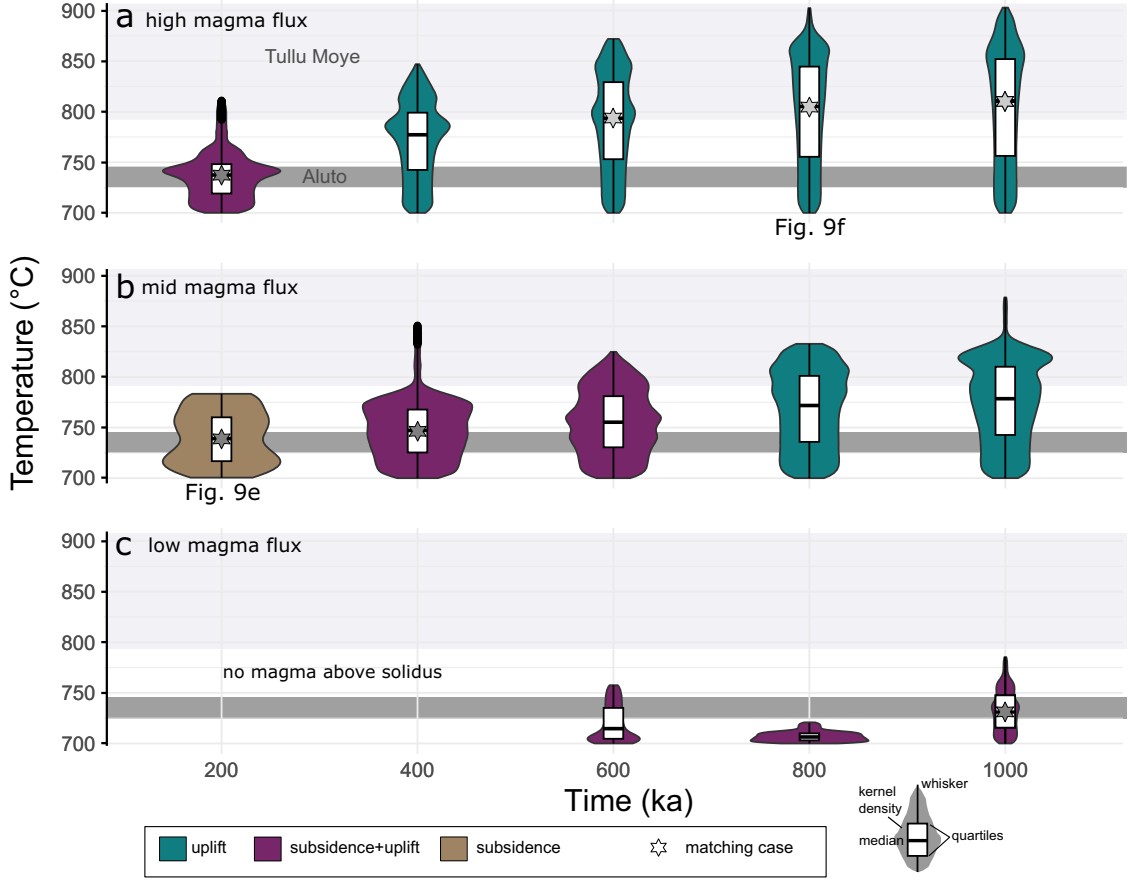

**Fig. 10 | Violin-boxplots of modeled temperature distributions of magmatic systems above the solidus temperature.** The solid bars in each panel represent the range of temperatures reconstructed for Tulu Moye (light gray) and Aluto (dark gray) as reconstructed from magnetotelluric and petrological modeling[61,64,65]. **a** High, **b** intermediate, and **c** low magma flux scenarios are shown. Color coding reflects the deformation pattern. Matches of average computed temperature distributions and reconstructed temperature ranges for Tulu Moye and Aluto are shown as gray stars.

200 ka) and another for Tulu Moye (high flux, 800 ka), both matching the observed temperature distributions (Fig. 10). Although multiple simulations could fit the temperature profiles for each volcano, the cases matching Tulu Moye consistently show monotonous uplift, leading to similar deformation time series. For Aluto, we selected a simulation depicting subsidence over timescales comparable to the observed deformation series, acknowledging that other simulations might show a higher frequency of uplift-subsidence cycles compared to what we observe at Aluto. Although our results do not rule out any contribution of hydrothermal processes[13] in shaping the deformation, we note that the source depth >5–8 km are difficult to reconcile with the imaged extent of hydrothermal alteration <2 km. We therefore propose that the differences in deformation regime for Tullo Moye and Aluto are primarily a reflection of the contrasting thermo-mechanical architecture of the magmatic plumbing systems.

## Implications for volcanology

The findings presented in this study carry significant practical implications for understanding volcanic unrest signals. Our simulations indicate that spatial deformation patterns can undergo time-dependent shifts when magma bodies interact with a heterogeneous thermal structure of the crust. Misinterpreting such changes in spatial deformation patterns resulting from viscoelastic creep of crustal rocks as indications of underground magma movement could result in erroneous forecasts. Furthermore, our model calculations reveal characteristic temporal patterns of surface deformation for cold and hot magmatic plumbing systems, validated by fully independent geophysical imaging and InSAR data. This implies that deformation cycles, which are widely observed at caldera volcanoes, may mirror the underlying thermo-mechanical architecture of the plumbing system. This is important as it offers the opportunity to use differences in

surface deformation time series to gain insights into large-scale magmatic system architecture, going beyond conventional deformation source inversion.

Recent studies highlight the vital role of viscoelastic processes in volcano deformation[25,27,29,32,43,44,68], and our findings stress the importance of considering these processes more broadly, especially in shallow and young magmatic systems. The thermal architecture of transcrustal magmatic systems is formed by the repeated injection of numerous magma batches over prolonged periods of time. Unlike viscoelastic shell models or geometries with stacked elastic and viscoelastic layers, such systems exhibit a continuum of elastic and viscoelastic materials. This suggests that even shallow deformation sources (e.g., <5 km depth), often assumed to behave elastically, may be significantly impacted by time-dependent rheology with heterogeneous expression in different spatial dimensions. Incorporating temperature-dependent viscoelastic processes into the geodetic inversion of volcano deformation signals is crucial for accurate monitoring. Neglecting these factors could lead to inaccuracies in source depth or volume calculations, highlighting the necessity of integrating them into analysis methods. Such integration could effectively be based on the results of geophysical imaging, such as magnetotelluric and seismic methods, providing information on the distribution of crustal temperatures beneath volcanic systems.

In addition to these practical aspects, our study holds further implications for fundamental volcano science. We show that volcanic surface deformation time series can reflect thermal and rheological architecture of magmatic plumbing systems, as governed by the long-term magmatic flux, the history of injection locations, and system lifespan. Given our model prediction that monotonous uplift and subsidence-uplift cycles correspond to hot and cold magmatic systems respectively, these differences could be used for first-order estimates of subsurface magmatic conditions, such as temperature distributions and potentially long-term magma fluxes, which are otherwise difficult to estimate.

A major challenge in this conceptual framework is that multiple trade-offs exist between different variables that impact surface deformation fields. However, estimates of the lifespan of magmatic systems, magma volatile contents, depth range of injection, and long-term magma input rates can be recovered through geochronological and petrological methods[69–71]. Integrating age determinations and petrological studies with thermal and deformation modeling has therefore great potential to improve our understanding of the architecture and dynamics of igneous plumbing systems. This is crucial because, while surface deformation is now routinely measured on a global scale, our understanding of crustal magma budgets and the thermal state of volcanic plumbing systems remains limited.

In summary, our findings underscore how the evolving thermal and rheological architecture of magmatic systems over their lifecycle shape short-term volcano deformation dynamics, with far-reaching implications for fundamental volcanology, hazard mitigation and volcano monitoring strategies.

## Methods

### Modeling long-term magma evolution

Numerical simulations were conducted to investigate the long-term thermal evolution of magmatic systems and the surrounding wall rocks. The simulations involved the pulsed injection of dikes and sills and were ran using the open-source software package 'Magma thermo-kinematics' (MTK), implemented in the high-level programming language Julia (Schmitt et al. 2023). A comprehensive description of the software, including its source code and benchmarking against previously published thermal models[69] of pulsed magma injection, is provided in ref. 70 and is available at https://github.com/boriskaus/MagmaThermoKinematics.jl. Here we summarize the general methodology, describe the setup and parameters used in our specific case.

The core of the simulation is based on solving the heat diffusion-advection equation using the finite difference method. The governing equation used in Magma-thermokinematics can be expressed as:

$$\rho c_p \left( \frac{\partial T}{\partial t} + \upsilon_j \frac{\partial T}{\partial x_j} \right) = \frac{\partial}{\partial x_j} \left( k \frac{\partial T}{\partial x_i} \right) + H_r + \rho Q_L \frac{\partial \theta_s}{\partial t} \tag{1}$$

Here, T represents temperature in Kelvin, υ denotes the host-rock velocity in response to the injection of new sills or dikes, t signifies time in seconds, $\theta_s$ stands for the solid fraction, ρ is the density, $c_p$ denotes heat capacity, k represents the thermal conductivity, $H_r$ is the radioactive heat source, and $Q_L$ is the latent heat. When magma intrudes, it displaces the host rock to make room for itself. To model this process, an injection routine based on an analytical solution is used that inserts the dike temperature into the temperature field and defines the velocity field needed to create a penny-shaped crack in an elastic half space. To address numerical stability issues arising from the time derivative of the solid fraction ($\theta_s$), MTK locally adjusts the heat capacity to incorporate latent heat release, thereby enhancing the stability of the solution. Furthermore, since material properties, such as melt fraction, are temperature-dependent, a non-linear iteration loop is employed.

The software discretizes the time derivative explicitly, using the following stable timestep criterion:

$$\frac{\min\left(\Delta x^2, \Delta y^2\right)}{\kappa} < 10^{-1} \tag{2}$$

where $\Delta x^2$ and $\Delta y^2$ are the grid spacings, and $\kappa$ is the thermal diffusivity ($\kappa = k/(\rho\, c_p)$). A staggered grid finite difference scheme has been employed to compute the spatial derivative, effectively addressing variations in thermal conductivity between temperature grid points. The implementation incorporates a temperature-dependent thermal conductivity[72].

Simulations were carried out in 2D geometry to track the spatial development of thermal heterogeneity within the crust. Two sets of geometries were employed to assess the impact of different domain sizes: (1) A geometry with a physical domain size of 20 × 20 km, discretized with 300 × 300 numerical nodes. (2) A larger geometry with an expanded horizontal physical domain size of 40 km and a depth of 20 km, discretized by 600 × 300 numerical nodes. Initial testing showed that these geometries provide similar results (Fig. S4). We thus focus in the following only on the smaller geometry case.

A generic melting parameterization, implemented in MTK based on ref. 71 was used for this study. In the used parameterization, melt fraction is calculated from the temperature in °C as:

$$\Psi = \frac{(800 - T)}{30} \tag{3}$$

$$\theta_{melt} = \frac{1}{(1 + e^{\Psi})} \tag{4}$$

with a liquidus-solidus range of 1000−700 °C, which is applicable for intermediate to silicic magma compositions. The initial temperature profile was determined by applying a linear geothermal gradient of 25 °C per kilometer. Flux-free boundary conditions were applied to the sides of the model. To explore various magma injection rate, we adjusted the fluxes by modifying the periodicity of sill and dike emplacement. In this study, we focus on three distinct scenarios labeled as follows: high magma flux case (4.7 × 10⁻⁴ km³/yr), intermediate flux (3.2 × 10⁻⁴ km³/yr), and low magma flux case (1.9 × 10⁻⁴ km³/yr). These magma fluxes have been chosen to capture a wide range of crustal thermal histories with incubation periods for magma reservoir growth ranging between ~200 ka (high flux case) to ~1000 ka (low flux case) (Fig. 2). The volume of dikes and sills was computed based on the width

(200 m) and length (1500 m) of injected dikes, assuming a penny-shaped geometry. Consequently, individual injections resulted in a volume of 0.94 km$^3$. Injection locations were randomly distributed over a height of 10 km and a width of 6 km relative to the center of the entire domain. Injection angles of dikes and sills were randomly varied from 80 to 100 and −10 to 10°, respectively. To assess the impact of random injection on the thermal evolution of the magmatic system, the intermediate flux scenario was repeated using a different randomized set of injection locations. All simulations ran over a total duration of 1 million years. The evolving temperature field was tracked at intervals of 200,000 years and subsequently utilized as input for crustal deformation modeling.

### Surface deformation modeling

To compute short-term surface deformation pattern at different stages of the evolutionary history of magmatic systems, we coupled the thermal output with thermo-mechanical models. Crustal deformation modeling was done using the structural mechanics and heat transfer modules of the finite element code COMSOL Multiphysics version 5.1. The model setup is based on the benchmark procedure for surface deformation developed in ref. 53. We used a 2D geometry, in order to determine the impact of spatial heterogeneity of the temperature field on the results, which is in not easily accounted for in the more commonly used axisymmetric model formulations. We note that the 2D geometry overestimates the magnitude of deformation compared to previous benchmark models[53], but this is less problematic as we are interested in general trends and differences between simulations at various evolutionary stages of the magmatic system rather than modeling deformation at a specific volcanic system.

Model geometries of either a 20 × 20 km or 20 × 40 km were used equivalent to the thermal simulations. First, the temperature field corresponding to a particular evolutionary stage from the thermal model was incorporated using an interpolation function. The geometry was then parameterized as a temperature-dependent viscoelastic rheology using a Standard Linear Solid material model with the following properties: Heat capacity of 1050 J/kg K, density of 2700 kg m$^{-3}$, thermal conductivity of 2 W/(m K), Bulk modulus of 13.3 × 10$^9$ N/m$^2$, and shear modulus of 8 × 10$^9$ N/m$^2$. In addition, a subset of calculations was carried out with generalized Maxwell rheology, using the same set of parameters (Fig. S5). For reference, a comprehensive list of parameters employed in the thermal and thermo-mechanical simulations is given in Table S1. Additionally, the modeling setup is illustrated in Fig. S5.

Viscosity was calculated using an Arrhenius law of the form:

$$\eta = A_d \, e^{\left(\frac{A_e}{(RT)}\right)} \tag{5}$$

where A$_d$ is the Dorn parameter of 10$^9$ Pa s, A$_E$ is the activation energy of 120,000 J/mol, R is the gas constant of 8.134 J/(mol K), and T is the temperature field from the thermal simulations. The relaxation time was calculated as:

$$\tau = \frac{\eta}{(G_{inst}\,\mu_1)} \tag{6}$$

where η is the viscosity, G$_{inst}$ is the instantaneous shear modulus, and μ$_1$ is the fraction (0.5) of the shear modulus of the viscoelastic arm.

Overpressure sources were modeled as ellipsoidal cavities with dimensions of 800 × 200 m at a depth of either 5, 10 or 15 km, unto which a constant boundary load of 1 MPa was applied if not stated otherwise (e.g., Figure 9; Figure. S6; Fig. S3). Given the impact of cavity size on surface deformation (e.g., refs. 43,44), we also explored smaller and larger source dimensions as shown in Fig. S7. Conceptually, we envision that the modeled overpressure sources represent either single intrusive bodies (e.g., shallow laccoliths), pressurized melt lenses in a

crystal mush, or recharge events into a mush zone without invoking magma mixing or mingling processes. It is important to note, however, that we do not explicitly model the intrusion process that leads to cavity formation. Consequently, the initial elastic response should not be interpreted as resulting from a specific igneous process. Instead, our focus is solely on the thermo-viscoelastic processes resulting from a pre-formed, overpressurized cavity. The surface was modeled as a free boundary, while roller boundaries were applied to the sides of the model domain, and a fixed constraint corresponding to the initial condition was applied to the bottom of the model domain. The model geometry was discretized with typically >31,000 triangular mesh elements.

We solved the temperature-dependent crustal deformation field in two steps following ref. 53. In a first step, the temperature field from the thermal modeling results, corresponding to a particular evolutionary stage (e.g., 400 ka of magmatism), was implemented through the heat transfer module in COMSOL Multiphysics. In a second step, crustal deformation in response to the overpressure source was solved for a duration of 10 years with step size of 6 month using the solid mechanics solver. We assume that the temperature field can be treated as quasi-stationary over the timespan of interest (10 years). The COMSOL.mph and thermal input files are available in a zenodo repository.

## Data availability

The specific MagmaThermoKinematics runfile used in this study, along with the temperature fields for high, mid, and low magma flux scenarios across various evolutionary stages of the magmatic system (0:200:1000 ka), as well as the COMSOL Multiphysics.mph input files, are available in the Zenodo repository at DOI: 10.5281/zenodo.14066709.

## Code availability

The source code for the MagmaThermoKinematics software can be accessed on Github: https://github.com/boriskaus/MagmaThermo Kinematics.jl. or under the Zenodo https://doi.org/10.5281/zenodo.7985281. COMSOL Multiphysics.mph input files are available in the Zenodo repository at DOI: 10.5281/zenodo.14066709.

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

## Acknowledgements

This project was funded by the European Union (ERC, MAST, Grant agreement ID: 101003173, J.B.), and the NERC-BGS Centre for the Observation and Modeling of Earthquakes Volcanoes and Tectonics (COMET). We would like to thank Jo Gottsmann help with COMSOL Multiphysics and Edna Dualeh for providing deformation time series data for Tulu Moye. Steve Sparks is thanked for discussion of the results at an early stage of this project. We are grateful to Friedemann Samrock for kindly giving us access to an editable version of figure on magneto-telluric imaging of Aluto and Tulu Moye. Boris Kaus is thanked for prompt assistance with queries regarding the 'MagmaThermoKinematics' code.

## Author contributions

Conceptualization: G.W., J.B., and C.A. Methodology and Software: G.W. Formal analysis and Investigation: G.W. and J.B. Writing: G.W., J.B., and C.A. Funding acquisition: J.B.

## Competing interests

The authors declare no competing interests.
