## [Transparent Peer Review file · Nature Communications]

Distinct patterns of volcano deformation for hot and cold magmatic systems

Corresponding Author: Dr Gregor Weber

Version 0:

Reviewer comments:

Reviewer #1

(Remarks to the Author)

In their paper "Distinct patterns of volcano deformation for hot and cold magmatic systems" Wenner et al. develop a long term (1 Ma) thermal model of crustal magma system establishment (using low, mid and high background magmatic fluxes) and evolution, based on random dike and sill injections into a predefined 2D region to assess temperature, melt-fraction and viscosity of the medium over time. These results are used as background properties into which ellipsoidal pressure sources and 5, 10, 15 km are embedded (different model, COMSOL), to model (vertical) displacements due to an impulse pressurization and the following visco-elastic response of this heterogeneous medium observed over a 10-year timespan. Depending on stage in the evolution (200-1000 ka) and the resulting system temperature that also depends on magmatic flux, the authors find substantially different deformation responses that range from expected post-eruptive exponentially decaying uplift for hot systems to subsidence and uplift episodes for colder systems. To explain this, they invoke viscous creep and its interaction with (a) an evenly hot surrounding material, which gives an equal viscoelastic response in all directions, and (b) a system that is hotter below than above, resulting in downwards viscous creep first and subsidence. They are able to reproduce the deformation behaviour at two systems in the East African Rift (Aluto and Tullu Moye) using MT observations to constrain the background temperatures and inferences about the evolution stages of the systems / long term magma flux.

This paper puts forward an interesting mechanism to further our understanding of magmatic systems and the sometimes puzzling deformation fields we are concerned with. They rightfully point out that they report a behavior that has not been captured in analytical models and support through their results the importance of considering heterogeneously warmed crust and the importance of where injections occur inside the background temperature field. I guess a main issue with their model is the dependence on both background temperature and long-term magmatic flux, which are both difficult to constrain and result in overlapping scenarios that can create different deformation fields (see their East Africa example).

I believe, this concept is important to convey to the broader community and will be of great interest, which warrants publication in Nature Communications in my view. My comments below probably place the necessary revisions somewhere between minor and major, somewhat depending on what the figures show etc.

1) There's a lot to consider in this paper: magma flux, evolutionary stage of the magmatic system, depth of the injection, or generally location of injection into the background temperature field. Clearly, a short form paper and touch on only so many of these aspects, but I believe the paper would benefit from some reconsiderations of cases that are considered: It seems that endmember cases might be best suited to convey the concepts: high and low flux, shallow and deep injections (and perhaps further consideration of inside the hot area and adjacent to it). As the paper is written, it is hard to figure out which change results in what deformation field. Figure 6 is a good example: the authors change the location of the injection, but also the background temperature field AND the evolution stage of the system AND the background flux. They title this figure "Influence of thermal heterogeneity on spatial surface displacement" ... it seems this might be well achieved by using the same, somewhat asymmetric background temperature field and injecting centrally and adjacent to demonstrate its importance relative to the location of the magma recharge. It might well be that I am missing something here, that in itself might be cause to restructure a bit.

2) To this point - Figure 4 shows the importance of the injection history (dotted line) on the deformation response. I assume

this trades off with the injection location? While I agree that the dotted line shows similar behavior for the intermediate case, it seems that the subsidence nature is lost sooner (compare solid and dotted orange lines for 5 km case for 800 and 1000 ka). This brings the same flux closer to the high flux scenario, solely based on injection location, I presume, further complicating the tradeoffs between parameters mentioned above. It seems the authors should point this out more explicitly than is currently done in L135-140, which doesn't mention the later stages of magma system evolution.

3) If the authors retain the intermediate flux case, it should be retained in Figure 5, too.

4) Figure 7: it would be good to show the two contrasting cases here (high/low flux).

5) For the East Africa example, I am not able to follow why they chose the respective scenarios. The text is somewhat vague and does not specify clearly (maybe I couldn't find it) why the 200 and 600 ka and cold / hot scenarios are chosen (shown in figure 8) instead of the low and high flux after 1 Ma, or intermediate flux after 400 ka and 1Ma, which are mentioned in the text. Perhaps best would be a comparison of deformation predictions based on several models to demonstrate the impact of these choices (but maybe the deformation histories will not be matched as well?). Presumably, Aluro went through 2 injections during that time period (Y1, Y4)? Can't find that specified in the text. I would also be really curious if they've found a data set that shows the migration of the uplift center as suggested in figure 6b,d,f.

6) I am bit puzzled by the limitations on "implications for volcano monitoring." While this is an important aspect, it seems that this work also has important implications on basic volcano science. Could the authors show how we could use different deformation fields to infer heat / melt fraction distributions (even to first order), or long-term magmatic flux for these systems? That is, can we learn new things beyond the classic elastic homogeneous isotropic half-space knowledge we commonly get from deformation fields with their insights? If so (and I think we can, but there are substantial trade-offs, see above), this should be pointed out in some conceptual sense as it would increase the paper's impact.

Minor issues:

Line 9-10: why just forecasting?

Intro: might perhaps be worth including non InSAR references, given the long timeseries that we are accumulating at some GNSS monitored volcanoes, particularly highlighting these processes?

L48: numerical model_s_

equation 2: is there a < missing?

503: was -> were

sometimes Pas or Pa s ... use the latter

L158-161: could be observable with GNSS (or tilt) if we knew what we're looking for...

Figure 4: dotted orange line really hard to see...

L209: add dot after "rates"

L233 - 238: should be observable in data - seen anywhere?

Reviewer #2

(Remarks to the Author)

This manuscript presents interesting results that highlight the role of thermo-elastic deformation. I find of particular noteworthiness the results around the influence of thermal heterogeneity and how this produced corresponding heterogeneous viscoelastic deformation responses. The work will be of significance to anyone in the field of volcano deformation.

I see no major flaws in the method or analysis, and comments/suggestions for improvement can be found in the attached file. These are minor and involve three main themes.

1. A "toning" down of some of the claims in the introduction. This work included novel aspects, but there have been several studies with comparable elements that have looked at non-static thermally controlled deformation models of volcanic systems (a good few of which are cited by the authors).

2. I got there in the end, but I did find figure 4 initially difficult to understand, I may have not read thoroughly enough the opening few pages, or not realised the importance, but I think this needs work to make sure that the results are supported by the description of the methodology, a really quick win would be referring back to Figure 1 when discussing Figure 4.

3. I have no issue with the underlying methodology, it is appropriate and understandable. I would however like to see more

commentary on the creation of the 800m x 200m pressurised cavity, or specifically more commentary on what it is not. I may be being oversensitive to this, but the manuscript is presented as if the results being envisaged as created from a distinct intrusion or equivalent, after a given period of magmatic activity. However, the result does not really model the intrusion of anything, as the cavity is pre-formed in the model mesh and then pressurised. Any comments therefore on the initial elastic response cannot be overly interpreted, and the appropriateness of it representing an intrusion is also limited. The strength, as mentioned previously is looking at the viscoelastic response, and the authors already allude to this in lines 147-148. I would like to see it more explicitly expressed that the initial elastic response does not represent any expected volcanic process, while emphasising the observations on the thermoelastic response.

These are minor points and mainly clarifications, I have made a few suggestions around cleaning up the text in the introduction, otherwise I enjoyed reading the manuscript which details an interesting and useful piece of work.

Reviewer #3

(Remarks to the Author)

This manuscript takes a creative approach to the problem of reconciling different timescales involved in volcano deformation. The authors model 100s kyr thermal evolution in a crustal section, which then provides initial conditions and defines static but spatially heterogeneous material properties in a 1-10 year viscoelastic model of a single spheroidal intrusion. The approach merges notions of thermal conditioning and long-term intrusive flux with the cavity-type descriptions of magma chambers used to model active volcano deformation.

The numerical modeling is 2D, a mash-up of a commercial, non-open source FEM code (COMSOL) and an open source finite difference code. The authors identify some interesting and somewhat non-intuitive results that suggest a role for thermal maturity - and by extension the history of magmatic flux - in volcano deformation timeseries, including scenarios for which intrusion results in subsidence due to viscoelastic effects and cyclic behavior. The authors apply these results in a qualitative way to data from volcanoes in the East African Rift.

I support publication – depending how the authors address the comment listed in the first below. But overall, it would be nice to see a more thorough parameter exploration to argue the main points convincingly. The authors could also do a better job justifying various assumptions in the model, as they are not unique and future work will benefit from a clearer exposition.

Before I believe the results, I'd like to know whether the subsidence you observe in models is simply a result of inconsistent model initial conditions.

You are modeling long-term assembly in a kinematic way, but the system has mechanical memory and stresses associated with emplacement of intrusions have long-range influence. Rucker et al., (2022) found that steady state cavity size varied with forcing period because viscoelastic creep is frequency dependent. This implies that the “size” of a magma chamber can't simply be imposed kinematically in time-dependent viscoelastic problems... it's a model result. My impression is that most viscoelastic chamber models do not address mechanical initial conditions. But chamber volume impacts surface deformation, so this impacts the significance of your results. Similar issues relating to pre-stress have been long recognized in earthquake models (e.g., Day, 1982). We're sort of behind the curve in volcanology.

I'm also curious whether you'd see subsidence even in the absence of sill emplacement... since you're coupling two different codes (two entirely different numerical methods!), self-consistency must be demonstrated. It could be that one needs to derive initial equilibrium mechanical conditions in an iterative manner (similar to what is done with gravitational loading from topography in elastic problems, e.g., Cianetti et al., 2012).

Other comments:

You use Standard Linear Solid rheology but it's a little unclear how you're deriving a single relaxation timescale (e.g., line 159). I am actually a bit skeptical of the need to use SLS rheology here despite several recent papers that advocate for it. SLS seems more a way to get sensible deformations from a physically implausible model setup (constant step chamber pressure increase implies constant magma flux into the chamber, does it not? Not even an oversimplified magmatic approach to mass influx implies that...) rather than anything about crustal rheology. Spatially variable Arrhenius temperature dependence and time variable forcing in a Maxwell model will trade off with band-limited effects from the SLS model. Shouldn't we use the simplest model until real data dictate otherwise? Regardless, this all needs to be tightened up, explicit rheologic and mass balance assumptions stated etc.

And it's not simply the relaxation time that governs deformation in viscoelastic systems as described on line 159. It's the ratio of relaxation time to forcing timescale (the Deborah number, for Maxwell rheology) that determines whether the surface deformation reflects elastic or viscous response (Rucker et al., 2022; Liao et al. 2023). I think you could do a more convincing job with this problem if you developed the appropriate nondimensional numbers and presented your results in this way. For example, it would be great to say something more concrete about how the spatial structure of the thermal field impacts the mechanical evolution (paragraph starting in 223). What sets the lateral extent of the chamber or the thermal preconditioning here? It plays a big role. Then I'd seek to explicitly tie the influx rate to the thermal preconditioning and then to the deformation. The latter is our main observation, and it would be nice to know whether anything unique is encoded there. Could be the answer is no.

Its worth noting that Liao et al (2023) has essentially resolved the dichotomy between older “shell” models and spatially variable thermoviscoelastic models of magma chamber deformation. The “shell” can be precisely defined and derived analytically for each frequency component of forcing, and one can associate the “shell” spatial extent with a Deborah number contour (even for step function pressure as modeled here). That’s a fairly big result in my opinion.

Finally, perhaps this is more of a philosophical comment, but I’m struck by some phenomenologic similarities between this manuscript and that in Karlstrom et al., (2010). For example, Fig 4 in that paper doesn’t look so different to me than Fig 3 in the present manuscript (appropriately blurring the eyes). Certainly the 2D stochastic approach to long-term recharge and notions of thermal pre-conditioning as governing short-term dynamics are similar. There are so many benefits to the modeling framework developed here of course, but I suspect that the basic phenomenology of your results could be (and to some extent already have been) more simply demonstrated.

Great, provocative work regardless.

Leif Karlstrom

References

- Cianetti, S., C. Giunchi, and E. Casarotti (2012), Volcanic deformation and flank instability due to magmatic sources and frictional rheology: The case of Mount Etna, *Geophys. J. Int.*, 191, 939–953, doi:10.1111/j.1365-246X.2012.05689.x.
- Day, S. M. (1982). Three-dimensional simulation of spontaneous rupture: the effect of nonuniform prestress. *Bulletin of the Seismological Society of America*, 72(6A), 1881-1902.
- Karlstrom, Leif, Josef Dufek, and Michael Manga. "Magma chamber stability in arc and continental crust." *Journal of Volcanology and Geothermal Research* 190, no. 3-4 (2010): 249-270.
- Liao, Yang, Leif Karlstrom, and Brittany A. Erickson. "History-Dependent Volcanic Ground Deformation From Broad-Spectrum Viscoelastic Rheology Around Magma Reservoirs." *Geophysical Research Letters* 50, no. 1 (2023): e2022GL101172.
- Rucker, C., B. Lee, J. Gopalakrishnan, B. A. Erickson, and L. Karlstrom. "A Computational Framework for Characterizing Viscoelastic Effects Surrounding a Buried Magma Reservoir."

Version 1:

Reviewer comments:

Reviewer #1

(Remarks to the Author)

The changes in response to my first round of comments are overall very satisfying. Thanks to the authors for thorough revisions!

I think, however, that I lost a piece in my prior review transferring it to the online system. I think it's still worth including here. The manuscript might benefit from some discussion related to this, especially given the new text around lines 486-491.

The current claim that uniform uplift vs. uplift subsidence cycles are due to temperature / hot vs cold magmatic systems may be a bit strong. There have been other papers investigating various mechanisms unrelated to viscoelasticity, some of which already cited. More related to this work, though, Block et al. (2023), while using static rheology, suggested that complex spatial and temporal surface deformation pattern can be created over long time scales just by pressure variations over time (which are part of the model here too, at least in creating the subsurface structure) into a magma reservoir embedded in a viscoelastic region. They also demonstrate time dependent shifts in surface deformation, especially a phase lag between "central" and "shoulder" regions that they attribute to viscosity, and propose this as an alternative to the sombrero pattern observed at mid-crustal magma bodies. This is even for a relatively young (10kyrs?), but deeper system, the Socorro Magma Body, which seems to be at odds with the findings in this current paper. Perhaps that's due to the focus on the region of maximum uplift by Weber et al. I am not suggesting that the authors explore all temporal variations in their modeled signal - that's for later work that this paper opens up. But it seems that some more caution in the interpretation of their results is needed - clearly interpretations of surface deformation fields (often discontinuous and even missing regions with max deformation) will depend heavily on our prior knowledge of the system, and while we could interpret change in deformation over time as hot v. cold it could also be other reasons, such as viscous relaxation due to pressure reduction.

Once this aspect is addressed, I think this will make a fine contribution. I don't need to see the paper again.

Block, G.A., M. Roy, E. Graves, R. Grapenthin (2023), Pressurizing Magma Within Heterogeneous Crust: A Case Study at the Socorro Magma Body, New Mexico, USA, *GRL*, vol 50(20), e2023GL105689

Reviewer #2

(Remarks to the Author)

Thank you for taking the time to respond to my comments in a way which makes the manuscript, in my opinion, considerably easier to follow and understand.

My positive comments from the initial review still stand, and with the added clarity following revision, I think this is a very good piece of work and have no concerns with recommending publication in its revised form.

Reviewer #3

(Remarks to the Author)

I appreciate the considerate responses of the authors to reviewer comments, and feel the manuscript is much improved. Its a nice study.

I'm still hung up on your justification of initial conditions, and subsequent discussion of the origin of subsidence. I don't think you addressed that concern completely.

Its not simply that "the location and size of the source would be controlled by the mechanical conditions and history-dependent stress field" - the nature of the deformation response depends heavily on this. As you state, subsidence following cavity pressurization only occurs under certain influx conditions (which set the spatial distribution of temperature hence viscosity). Could you better predict when downflow will occur based on simple analysis of the viscous force balance? Given the neglect of gravity (which may be tenuous in the case of viscous flow... certainly justified for elastic response), seems like the existence of viscous response in downward direction could be straightforwardly predicted based on magnitude of overpressure versus viscosity structure.

But then you are assuming a uniform overpressure and zero initial stress. What about the magma injection that (presumably) is your preferred origin for intrusion pressurization? That must have itself flowed down pressure gradient into the chamber. Would that deeper pressure source balance the downward flow of hot host rock? Thats the type of "pre-stress" I was referring to. I think you need to more clearly contextualize the assumptions you're making about magma mass transport, even if thats not modelled explicitly. I'm guessing that this subsidence behavior is an end member at best.

Version 2:

Reviewer comments:

Reviewer #3

(Remarks to the Author)

The authors have addressed my concerns sufficiently and I'm happy to see this published.

Reviewer #1:

In their paper "Distinct patterns of volcano deformation for hot and cold magmatic systems" Wenner et al. develop a long term (1 Ma) thermal model of crustal magma system establishment (using low, mid and high background magmatic fluxes) and evolution, based on random dike and sill injections into a predefined 2D region to assess temperature, melt-fraction and viscosity of the medium over time. These results are used as background properties into which ellipsoidal pressure sources and 5,10,15 km are embedded (different model, COMSOL), to model (vertical) displacements due to an impulse pressurization and the following visco-elastic response of this heterogeneous medium observed over a 10-year timespan. Depending on stage in the evolution (200-1000 ka) and the resulting system temperature that also depends on magmatic flux, the authors find substantially different deformation responses that range from expected post-eruptive exponentially decaying uplift for hot systems to subsidence and uplift episodes for colder systems. To explain this, they invoke viscous creep and its interaction with (a) an evenly hot surrounding material, which gives an equal viscoelastic response in all directions, and (b) a system that is hotter below than above, resulting in downwards viscous creep first and subsidence. They are able to reproduce the deformation behaviour at two systems in the East African Rift (Aluto and Tullu Moyo) using MT observations to constrain the background temperatures and inferences about the evolution stages of the systems / long term magma flux.

This paper puts forward an interesting mechanism to further our understanding of magmatic systems and the sometimes puzzling deformation fields we are concerned with. They rightfully point out that they report a behavior that has not been captured in analytical models and support through their results the importance of considering heterogeneously warmed crust and the importance of where injections occur inside the background temperature field. I guess a main issue with their model is the dependence on both background temperature and long-term magmatic flux, which are both difficult to constrain and result in overlapping scenarios that can create different deformation fields (see their East Africa example).

I believe, this concept is important to convey to the broader community and will be of great interest, which warrants publication in Nature Communications in my view. My comments below probably place the necessary revisions somewhere between minor and major, somewhat depending on what the figures show etc.

1) There's a lot to consider in this paper: magma flux, evolutionary stage of the magmatic system, depth of the injection, or generally location of injection into the background temperature field. Clearly, a short form paper and touch on only so many of these aspects, but I believe the paper would benefit from some reconsiderations of cases that are considered: It seems that endmember cases might be best suited to convey the concepts: high and low flux, shallow and deep injections (and perhaps further consideration of inside the hot area and adjacent to it). As the paper is written, it is hard to figure out which change results in what deformation field. Figure 6 is a good example: the authors change the location of the injection, but also the background temperature field AND the evolution stage of the system AND the background flux. They title this figure "Influence of thermal heterogeneity on spatial surface displacement" ... it seems this might be well achieved by using the same, somewhat asymmetric background temperature field and injecting centrally and adjacent to demonstrate its importance relative to the location of the magma recharge. It might well be that I am missing something here, that in itself might be cause to restructure a bit.

We appreciate the constructive suggestions regarding the clarity of our manuscript and made several revisions to address these points.

Firstly, we have revised Fig. 6 to better illustrate the impact of thermal heterogeneity while keeping other variables constant. In the revised version, we show two scenarios where injections are made centrally and adjacent to the hot area, using the same long-term magma flux and evolutionary stage of the system.

Secondly, we have revised the text to improve clarity regarding the impacts of varying injection locations, background temperature fields, evolutionary stages of the system, and magma fluxes. Specifically, we have updated the discussion in Lines 269-277 to provide a more explicit explanation of how each variable influences the resulting deformation field.

Given the feedback from other reviewers, particularly Reviewer 3's request for an expanded parameter space exploration, we have retained the intermediate flux scenario. However, we have moved any additional parameter explorations beyond those directly relevant to the central findings into the supplementary materials.

2) To this point - Figure 4 shows the importance of the injection history (dotted line) on the deformation response. I assume this trades off with the injection location? While I agree that the dotted line shows similar behavior for the intermediate case, it seems that the subsidence nature is lost sooner (compare solid and dotted orange lines for 5 km case for 800 and 1000 ka). This brings the same flux closer to the high flux scenario, solely based on injection location, I presume, further complicating the tradeoffs between parameters mentioned above. It seems the authors should point this out more explicitly than is currently done in L135-140, which doesn't mention the later stages of magma system evolution.

We agree that the effect of injection site history on surface deformation could be more clearly highlighted in the manuscript. The relevant section of the text has been revised and expanded:

“As illustrated by a repeat run of the intermediate flux scenario (dashed lines in Figs. 4, 5b), the magnitude and temporal evolution of surface deformation can be influenced by the long-term history of injection locations, which are randomized in our modelling framework. Although the overall temporal trends are similar, the spatial injection history can significantly affect the magnitude and duration of viscoelastic subsidence and uplift patterns. More spatially focused magma injection can lead to higher local temperatures and, consequently, greater magnitudes of viscoelastic deformation (e.g., intermediate flux 800 and 1000 ka repeat runs; Fig. 5b). Therefore, the history of spatial injection locations may be an important parameter in explaining the diversity of deformation signals that occur globally.” Lines 234-242.

3) If the authors retain the intermediate flux case, it should be retained in Figure 5, too.

The intermediate flux scenario has been added to the figure.

4) Figure 7: it would be good to show the two contrasting cases here (high/low flux).

The figure has been modified as suggested.

5) For the East Africa example, I am not able to follow why they chose the respective scenarios. The text is somewhat vague and does not specify clearly (maybe I couldn't find it) why the 200 and 600 ka and cold / hot scenarios are chosen (shown in figure 8) instead of the low and high flux after 1 Ma, or intermediate flux after 400 ka and 1Ma, which are mentioned in the text. Perhaps best would be a comparison of deformation predictions based on several models to demonstrate the impact of these choices (but maybe the deformation histories will not be matched as well?). Presumably, Aluro went through 2 injections during that time period (Y1, Y4)? Can't find that specified in the text. I would also be really curious if they've found a data set that shows the migration of the uplift center as suggested in figure 6b,d,f.

The manuscript was not sufficiently clear in pointing out why we chose specific simulations for comparison with the natural data. We now state in detail that this comparison is based on matching of temperature distributions in the model and published temperature estimates for the two volcanoes (Lines 382-397). To ease the comparison, a new Fig. 9 and Table S1 has been added showing temperature distributions and summary statistics of modelled magmatic temperature distributions in comparison to the published estimates for Aluto and Tullu Moye.

The specific choice of simulations which are shown in Fig. 8 is now discussed in Lines 441-448:

“In Fig. 8, we present two simulations: one for Aluto (intermediate flux, 200 ka) and another for Tullu Moye (high flux, 800 ka), both matching the observed temperature distributions (Fig. 9). Although multiple simulations could fit the temperature profiles for each volcano, the cases matching Tullu Moye consistently show monotonous uplift, leading to similar deformation time series. For Aluto, we selected a simulation depicting subsidence over timescales comparable to the observed deformation series, acknowledging that other simulations might show a higher frequency of uplift-subsidence cycles compared to what we observe at Aluto.”

With regards to modelled injections for Aluto: We mention in Line 411 that Aluto is modelled with 2 consecutive injection pulses. These are also shown in supplementary Fig. 3.

Regarding the migration of the uplift centre, we have checked at some possible candidates and see hints of a migration in the dataset at Tullu Moye. However, the quality of the data is not good enough to be included here - there are data gaps due to vegetation, which means the results are not conclusive. We will keep looking and if we do find something, it would be an excellent topic for a follow-up paper.

6) I am bit puzzled by the limitations on "implications for volcano monitoring." While this is an important aspect, it seems that this work also has important implications on basic volcano science. Could the authors show how we could use different deformation fields to infer heat / melt fraction distributions (even to first order), or long-term magmatic flux for these systems? That is, can we learn new things beyond the classic elastic homogeneous isotropic half-space knowledge we commonly get from deformation fields with their insights? If so (and I think we can, but there are substantial trade-offs, see above), this should be pointed out in some

conceptual sense as it would increase the paper's impact.

Thank you for this remark. A conceptual discussion of how the findings of our study could be used to infer magmatic conditions has been added:

“In addition to these practical aspects, our study holds further implications for fundamental volcano science. We show that volcanic surface deformation time series can reflect thermal and rheological architecture of magmatic plumbing systems, as governed by the long-term magmatic flux, the history of injection locations, and system lifespan. Given our model prediction that monotonous uplift and subsidence-uplift cycles correspond to hot and cold magmatic systems respectively, these differences could be used for first-order estimates of subsurface magmatic conditions, such as temperature distributions and potentially long-term magma fluxes, which are otherwise difficult to estimate.

A major challenge in this conceptual framework is that multiple trade-offs exist between different variables that impact surface deformation fields. However, estimates of the lifespan of magmatic systems, magma volatile contents, depth range of injection, and long-term magma input rates can be recovered through geochronological and petrological methods^{64,65,67}. Integrating age determinations and petrological studies with thermal and deformation modeling has therefore great potential to improve our understanding of the architecture and dynamics of igneous plumbing systems. This is crucial because, while surface deformation is now routinely measured on a global scale, our understanding of crustal magma budgets and the thermal state of volcanic plumbing systems remains limited.” Lines 484-500

Minor issues:

Line 9-10: why just forecasting?

Added “... magma dynamics ...”.

Intro: might perhaps be worth including non InSAR references, given the long timeseries that we are accumulating at some GNSS monitored volcanoes, particularly highlighting these processes?

We did include a wide range of GNSS studies in the introduction, including references 8, 10-11, 14-15.

L48: numerical model_s_

Thanks for spotting this. Corrected.

equation 2: is there a < missing?

Yes, this was missing. It has been added.

503: was -> were

Corrected.

sometimes Pas or Pa s ... use the latter

This has been corrected throughout the manuscript.

L158-161: could be observable with GNSS (or tilt) if we knew what we're looking for...

The sentence has been modified to “..., rendering them unlikely to be observable only with very high temporal resolution methods”.

Figure 4: dotted orange line really hard to see...

The colour has been changed to brown to increase visibility.

L209: add dot after "rates"

Added. Thanks.

L233 - 238: should be observable in data - seen anywhere?

This comment has been addressed above.

Many thanks for the comments.

Comments by Reviewer #2

This manuscript presents interesting results that highlight the role of thermos-elastic

deformation. I find of particular noteworthiness the results around the influence of thermal heterogeneity and how this produced corresponding heterogeneous viscoelastic deformation responses. The work will be of significance to anyone in the field of volcano deformation.

I see no major flaws in the method or analysis, and comments/suggestions for improvement can be found in the attached file. These are minor and involve three main themes.

1. A “toning” down of some of the claims in the introduction. This work included novel aspects, but there have been several studies with comparable elements that have looked at non-static thermally controlled deformation models of volcanic systems (a good few of which are cited by the authors).

We acknowledge that there are indeed several published models employing viscoelastic rheologies over timescales commonly used in volcano monitoring (e.g. Hickey et al. 2016) as correctly pointed out by the reviewer. However, these studies consider the temperature field and hence viscoelastic rheology to be static in time. We would therefore like to emphasize that our study addresses a different aspect of non-static temperature fields and therefore rheology by focusing on much longer timescales relevant to magmatic system construction (~100-1000 ka), which we believe has not been explored in the existing literature.

To address the reviewer's concern, we have rephrased the relevant section of the introduction to better clarify the novelty of our study. The revised text now reads:

“Given the temperature-dependence, the viscoelastic response of rocks subjected to strain is largely governed by the long-term thermo-mechanical history of the crust^{39,69}. While these timescales are much longer than those typically invoked in surface deformation modelling, we anticipate that magmatic systems undergo different styles of ground deformation at various stages of their lifecycle. This temporal evolution of crustal rheology may be essential in explaining the great diversity of spatial and temporal deformation behavior that is observed at different volcanoes worldwide⁴⁰. Yet, even state-of-the-art thermally controlled models of volcano deformation consider the temperature field and hence viscoelastic rheology to be static, often relying on simplified geometries of magma plumbing systems^{24,28}. This is the first attempt to reconcile the vastly different timescales of ground movements and magmatic system evolution to explore these dependencies in space and time.” Lines 55-65

2. I got there in the end, but I did find figure 4 initially difficult to understand, I may have not read thoroughly enough the opening few pages, or not realised the importance, but I think this needs work to make sure that the results are supported by the description of the methodology, a really quick win would be referring back to Figure 1 when discussing Figure 4

Thank you for pointing this out. To facilitate the interpretation of Figure 4, we have added a clarification that refers back to Figure 1. The revised text now reads:

“We model short-term deformation patterns over a time span of 10 years for overpressure sources at 5, 10 or 15 km depth and at different long-term evolutionary stages of magmatic systems (i.e. from 0 to 1 Ma in 200 ka increments; cf. Fig. 1).”

Additionally, the reference to Figure 1 has been included in the caption of Figure 4 to enhance clarity and support the description of the methodology.

3. I have no issue with the underlying methodology, it is appropriate and understandable. I would however like to see more commentary on the creation of the 800m x 200m pressurised cavity, or specifically more commentary on what it is not. I may be being oversensitive to this, but the manuscript is presented as if the results being envisaged as created from a distinct intrusion or equivalent, after a given period of magmatic activity. However, the result does not really model the intrusion of anything, as the cavity is pre-formed in the model mesh and then pressurised. Any comments therefore on the initial elastic response cannot be overly interpreted, and the appropriateness of if representing an intrusion is also limited. The strength, as mentioned previously is looking at the viscoelastic response, and the authors already allude to this in lines 147-148. I would like to see it more explicitly expressed that the initial elastic response does not represent any expected volcanic process, while emphasising the observations on the thermoelastic response.

We appreciate the feedback and agree with the concerns raised. To address this point, we have added a statement to the main text in Lines 193-199:

“Although the relaxation behaviour in our model is governed by distribution of viscosities rather than single values, we do not model the injection process itself. The step-change cavity pressurization does not reflect a specific igneous process and cannot be considered equivalent to a forcing timescale due to magma injection. Instead, our study focuses on the time-dependent thermo-viscoelastic behavior of crustal rocks in response to a pre-formed, pressurized cavity.”

and additional text in the methodology section (Lines 614-618):

"It is important to note, however, that we do not explicitly model the intrusion process that leads to cavity formation. Consequently, the initial elastic response should not be interpreted as resulting from a specific igneous process. Instead, our focus is solely on the thermo-viscoelastic processes resulting from a pre-formed, overpressurized cavity. "

Additionally, we have removed or revised any parts of the manuscript that implied the initial elastic response was interpreted as an intrusion process.

These are minor points and mainly clarifications, I have made a few suggestions around cleaning up the text in the introduction, otherwise I enjoyed reading the manuscript which details an interesting and useful piece of work.

Thank you for the comments. We appreciate the constructive feedback.

Reviewer #3 (Remarks to the Author):

This manuscript takes a creative approach to the problem of reconciling different timescales involved in volcano deformation. The authors model 100s kyr thermal evolution in a crustal section, which then provides initial conditions and defines static but spatially heterogeneous material properties in a 1-10 year viscoelastic model of a single spheroidal intrusion. The approach merges notions of thermal conditioning and long-term intrusive flux with the cavity-type descriptions of magma chambers used to model active volcano deformation.

The numerical modeling is 2D, a mash-up of a commercial, non-open source FEM code (COMSOL) and an open source finite difference code. The authors identify some interesting and somewhat non-intuitive results that suggest a role for thermal maturity - and by extension the history of magmatic flux - in volcano deformation timeseries, including scenarios for which intrusion results in subsidence due to viscoelastic effects and cyclic behavior. The authors apply these results in a qualitative way to data from volcanoes in the East African Rift.

I support publication – depending how the authors address the comment listed in the first below. But overall, it would be nice to see a more thorough parameter exploration to argue the main points convincingly. The authors could also do a better job justifying various assumptions in the model, as they are not unique and future work will benefit from a clearer exposition.

Before I believe the results, I'd like to know whether the subsidence you observe in models is simply a result of inconsistent model initial conditions.

You are modeling long-term assembly in a kinematic way, but the system has mechanical memory and stresses associated with emplacement of intrusions have long-range influence. Rucker et al., (2022) found that steady state cavity size varied with forcing period because viscoelastic creep is frequency dependent. This implies that the “size” of a magma chamber can't simply be imposed kinematically in time-dependent viscoelastic problems... it's a model result. My impression is that most viscoelastic chamber models do not address mechanical initial conditions. But chamber volume impacts surface deformation, so this impacts the significance of your results. Similar issues relating to pre-stress have been long recognized in earthquake models (e.g., Day, 1982). We're sortof behind the curve in volcanology.

Thank you for pointing out the findings of Rucker et al. (2022), which are certainly relevant to the problem, and we now reference in Lines 75, 193, 470 and 610. Clearly, the stress field and mechanical conditions are important factors to consider when modelling magma injection location and reservoir size, but in many cases, these will be overprinted by crustal heterogeneities which are not possible to model. We agree that a long-term goal should be a fully integrated model incorporating all these effects into a single modelling framework. However, the emphasis of this study is a new proof-of-concept to explore a new link rather than the development of a new modelling framework. Thus, we treat these confounding factors illustratively in our modelling approach at present. We have added: "In reality, the location and size of the source would be controlled by the mechanical conditions and history-dependent stress field^{63,68}, but for illustrative purposes, we use an oblate source situated at depths of either 5, 10, or 15 km and test a range of source sizes." to Lines 73-76.

As you correctly point out, reservoir size impacts surface deformation. We thus agree that it is important to test whether our subsidence trends are impacted by reservoir size. In the revised version of the manuscript, we present additional supplementary model

calculations with smaller and larger reservoir sizes (Fig. S7), showing that subsidence is also observed in these scenarios, reinforcing our main finding.

I'm also curious whether you'd see subsidence even in the absence of sill emplacement... since you're coupling two different codes (two entirely different numerical methods!), self-consistency must be demonstrated. It could be that one needs to derive initial equilibrium mechanical conditions in an iterative manner (similar to what is done with gravitational loading from topography in elastic problems, e.g., Cianetti et al., 2012).

As stated in L581-585, our thermo-mechanical model implements a temperature field into the benchmark approach of Hickey and Gottsmann (2014) that does not include gravity. Down-sagging of crustal rocks due to inconsistent mechanical equilibrium conditions is therefore not expected. Nevertheless, to test whether subsidence would occur without sill emplacement, we conducted simulations without applying any boundary load to the cavity. As expected, these simulations showed no displacement whatsoever.

Other comments:

You use Standard Linear Solid rheology but it's a little unclear how you're deriving a single relaxation timescale (e.g., line 159).

The single value relaxation timescales are just example calculations to orient the reader with regards to expected magnitudes of behaviour, bearing in mind the broad readership of Nature Communications. Of course, the model domain is fully parameterized in term so relaxation timescales as a function of spatially variable temperature. This is now stated more clearly in Lines 192-198.

I am actually a bit skeptical of the need to use SLS rheology here despite several recent papers that advocate for it. SLS seems more a way to get sensible deformations from a physically implausible model setup (constant step chamber pressure increase implies constant magma flux into the chamber, does it not? Not even an oversimplified magmatic approach to mass influx implies that...) rather than anything about crustal rheology. Spatially variable Arrhenius temperature dependence and time variable forcing in a Maxwell model will trade off with band-limited effects from the SLS model. Shouldn't we use the simplest model until real data dictate otherwise? Regardless, this all needs to be tightened up, explicit rheologic and mass balance assumptions stated etc.

Given that we are investigating general trends and pattern of surface deformation in an illustrative way, we would argue that the choice of Maxwell versus SLS rheological law is of secondary importance here. As you correctly point out, the question which rheology is more appropriate to real-world crustal rocks is currently discussed in the literature. Differences between these approaches have been investigated in detail elsewhere (e.g. Head et al. 2021).

Nonetheless, we agree that some comparison of our deformation time series for different rheologies may be informative and added a supplementary figure (Fig. S5) comparing our results with SLS and Maxwell rheology. This analysis shows that the overall pattern and trends of surface deformation are conserved.

We now state explicitly the assumptions of constant step change injection scenario (Lines 192-198 and Lines 614-618).

And it's not simply the relaxation time that governs deformation in viscoelastic systems as described on line 159. It's the ratio of relaxation time to forcing timescale (the Deborah number, for Maxwell rheology) that determines whether the surface deformation reflects elastic or viscous response (Rucker et al., 2022; Liao et al. 2023). I think you could do a more convincing job with this problem if you developed the appropriate nondimensional numbers and presented your results in this way. For example, it would be great to say something more concrete about how the spatial structure of the thermal field impacts the mechanical evolution (paragraph starting ln 223). What sets the lateral extent of the chamber or the thermal preconditioning here? It plays a big role. Then I'd seek to explicitly tie the influx rate to the thermal preconditioning and then to the deformation. The latter is our main observation, and it would be nice to know whether anything unique is encoded there. Could be the answer is no.

Thank you for this stimulating discussion that lays out a compelling pathway for further analysis. A brief discussion of Deborah Numbers has been added to Lines 192-198 of the revised manuscript.

We can see that casting the results in terms of Deborah numbers could provide additional insights. However, given that such calculations require knowledge of the forcing timescale, which in most modelled scenarios is treated as a constant cavity boundary load and therefore not representative of an injection timescale, this is beyond the immediate scope of this work. However, thank you for pointing this out. It would certainly be a good starting point for a follow-up study.

It's worth noting that Liao et al (2023) has essentially resolved the dichotomy between older "shell" models and spatially variable thermoviscoelastic models of magma chamber deformation. The "shell" can be precisely defined and derived analytically for each frequency component of forcing, and one can associate the "shell" spatial extent with a Deborah number contour (even for step function pressure as modeled here). That's a fairly big result in my opinion.

This is an interesting finding. We cite this study in Lines 75, 193, 470 and 610.

Finally, perhaps this is more of a philosophical comment, but I'm struck by some phenomenologic similarities between this manuscript and that in Karlstrom et al., (2010). For example, Fig 4 in that paper doesn't look so different to me than Fig 3 in the present manuscript (appropriately blurring the eyes). Certainly the 2D stochastic approach to long-term recharge and notions of thermal pre-conditioning as governing short-term dynamics are similar. There are so many benefits to the modeling framework developed here of course, but I suspect that the basic phenomenology of your results could be (and to some extent already have been) more simply demonstrated.

We added a reference to your work (Line 54-56)

Great, provocative work regardless.

Thank you for the comments.

Reviewer #1 (Remarks to the Author):

The changes in response to my first round of comments are overall very satisfying. Thanks to the authors for thorough revisions!

I think, however, that I lost a piece in my prior review transferring it to the online system. I think it's still worth including here. The manuscript might benefit from some discussion related to this, especially given the new text around lines 486-491.

The current claim that uniform uplift vs. uplift subsidence cycles are due to temperature / hot vs cold magmatic systems may be a bit strong. There have been other papers investigating various mechanisms unrelated to viscoelasticity, some of which already cited. More related to this work, though, Block et al. (2023), while using static rheology, suggested that complex spatial and temporal surface deformation pattern can be created over long time scales just by pressure variations over time (which are part of the model here too, at least in creating the subsurface structure) into a magma reservoir embedded in a viscoelastic region. They also demonstrate time dependent shifts in surface deformation, especially a phase lag between "central" and "shoulder" regions that they attribute to viscosity, and propose this as an alternative to the sombrero pattern observed at mid-crustal magma bodies. This is even for a relatively young (10kyrs?), but deeper system, the Socorro Magma Body, which seems to be at odds with the findings in this current paper. Perhaps that's due to the focus on the region of maximum uplift by Weber et al. I am not suggesting that the authors explore all temporal variations in their modeled signal - that's for later work that this paper opens up. But it seems that some more caution in the interpretation of their results is needed - clearly interpretations of surface deformation fields (often discontinuous and even missing regions with max deformation) will depend heavily on our prior knowledge of the system, and while we could interpret change in deformation over time as hot v. cold it could also be other reasons, such as viscous relaxation due to pressure reduction.

Thank you for sharing this discussion. We now acknowledge the mechanism proposed by Block et al. 2023 in Lines 44-45 of the revised manuscript and rephrased the text in the "Implications for volcanology" section more carefully in Lines 487 and 499.

Once this aspect is addressed, I think this will make a fine contribution. I don't need to see the paper again.

Block, G.A., M. Roy, E. Graves, R. Grapenthin (2023), Pressurizing Magma Within Heterogeneous Crust: A Case Study at the Socorro Magma Body, New Mexico, USA, GRL, vol 50(20), e2023GL105689

The Reference has been added.

Reviewer #2 (Remarks to the Author):

Thank you for taking the time to respond to my comments in a way which makes the manuscript, in my opinion, considerably easier to follow and understand.

My positive comments from the initial review still stand, and with the added clarity following revision, I think this is a very good piece of work and have no concerns with recommending publication in its revised form.

Thank you.

Reviewer #3 (Remarks to the Author):

I appreciate the considerate responses of the authors to reviewer comments, and feel the manuscript is much improved. Its a nice study.

Thank you.

I'm still hung up on your justification of initial conditions, and subsequent discussion of the origin of subsidence. I don't think you addressed that concern completely.

Its not simply that "the location and size of the source would be controlled by the mechanical conditions and history-dependent stress field" - the nature of the deformation response depends heavily on this. As you state, subsidence following cavity pressurization only occurs under certain influx conditions (which set the spatial distribution of temperature hence viscosity). Could you better predict when downflow will occur based on simple analysis of the viscous force balance? Given the neglect of gravity (which may be tenuous in the case of viscous flow... certainly justified for elastic response), seems like the existence of viscous response in downward direction could be straightforwardly predicted based on magnitude of overpressure versus viscosity structure.

As you correctly point out, the occurrence of subsidence can indeed be predicted based on the viscosity structure and the magnitude of overpressure. We have now incorporated an additional analysis to address this point. Specifically, we provide a new figure 8, which illustrates how the difference in relaxation timescales above and below the pressure source, calculated as η/G for the Standard Linear Solid (SLS) rheology, can be used to predict the conditions under which subsidence will occur. When the contrast in relaxation times is large, subsidence is likely to occur, whereas uplift tends to dominate when the difference is smaller. As you mention above, this could also be expressed in terms of overpressure, with the relaxation timescale given by η/σ , yielding a similar result.

We added the following discussion:

“The occurrence of either subsidence or uplift can be predicted by analysing the difference in relaxation timescales of crustal rocks, which is primarily governed by their viscosity structure. When there is a significant difference in relaxation timescales between the rocks above and below the pressure source, subsidence is likely, as the upper layer behaves elastically while viscous relaxation occurs in the lower layer (see Fig. 8). In contrast, when the difference is minimal, both layers undergo time-dependent deformation, resulting in net uplift. Thus, the relative difference in relaxation timescales, rather than absolute values, is the key factor controlling the deformation behaviour.”

Thank you for this comment. We believe it strengthened the manuscript.

But then you are assuming a uniform overpressure and zero initial stress. What about the magma injection that (presumably) is your preferred origin for intrusion pressurization? That must have itself flowed down pressure gradient into the chamber. Would that deeper pressure source balance the downward flow of hot host rock? Thats the type of "pre-stress" I was referring to. I think you need to more clearly contextualize the assumptions you're making about magma mass transport, even if thats not modelled explicitly. I'm guessing that this subsidence behavior is an end member at best.

Thank you for the clarification. To address your concern, we have significantly expanded the discussion surrounding the modelling assumptions. You raise a valid point that the context of these assumptions requires further elaboration, although this discussion must remain somewhat

speculative, as fully coupled magma injection models incorporating viscoelastic rheology are still beyond the current state-of-the-art. The following paragraph has been added:

“In this study, we do not assume any specific mechanism for pressure generation within the modelled cavity. However, two widely accepted processes for magma pressurization are the injection of fresh magma into a preexisting reservoir via a feeder dike and the accumulation of an exsolved volatile phase within the magma body⁷². For volatile-driven mechanisms, where pressure is generated internally, we do not expect these processes to directly influence the relaxation behaviour of the crustal rocks. However, magma transport through a feeder dike system could affect the stress field beneath the intrusion, potentially counteracting the downward-directed flow of hot host rocks. More complex models are needed to explore the interaction between magma injection and viscoelastic crustal rheology to determine whether this hypothetical effect would be significant. Nonetheless, petrologically constrained magma recharge timescales, typically ranging from days to months⁷², are shorter than the 10-year observation window modelled in this study. Therefore, our results are applicable to scenarios in which the injection process has terminated.”

The comments are very much appreciated. Thank you.